# TOWARD GENERALIZABILITY OF GRAPH-BASED IMPUTATION ON BIO-MEDICAL MISSING DATA

## ABSTRACT

Recent work on graph-based imputation methods for missing features has garnered significant attention, largely due to the effectiveness of their ability to aggregate and propagate information through graph structures. However, these methods generally assume that the graph structure is readily available and manually mask the original features to simulate the scenario of missing features. This set of assumptions narrows the applicability of such techniques to real-world *bio-medical tabular data*, where graph structure is not readily available and missing data is a prevalent issue, such as in cases involving confidential patient information. In light of this situation, and with the aim of enhancing generalizability in bio-medical domain, we propose GRASS that bridges the gap between recent graph-based imputation methods and real-world scenarios involving missing data in their initial states. Specifically, our approach begins with tabular data and employs a simple Multi-Layer Perceptron (MLP) layer to extract *feature gradient*, which serves as an additional resource for generating graph structures. Leveraging these gradients, we construct a graph from a feature (i.e., column) perspective and carry out column-wise feature propagation to impute missing values based on their similarity to other features. Once the feature matrix is imputed, we generate a second graph, but this time from a sample-oriented (i.e., row) perspective, which serves as the input for existing graph-based imputation models. We evaluate GRASS using real-world medical and bio-domain datasets, demonstrating their effectiveness and generalizability in handling versatile missing scenarios. The source code for our proposed method is available at `https://anonymous.4open.science/r/grass-iclr-41D5`.

## 1 INTRODUCTION

Missing data imputation (MDI) is a longstanding and pivotal research challenge across multiple disciplines (Schafer & Graham, 2002; Jerez et al., 2010). While traditional methods primarily rely on statistical techniques to leverage the distribution of non-missing data (Efron, 1994; Little & Rubin, 2019), recent advances in Graph Neural Networks (GNNs) (Kipf & Welling, 2016a; Hamilton et al., 2017; Veličković et al., 2017) have opened new avenues for MDI. Unlike conventional approaches that operate on tabular data, GNN-based methods employ a message-passing framework that effectively addresses the problem of missing features by aggregating information from neighboring samples. This proves especially useful in downstream tasks, often referred to as the ultimate goals of MDI (Taguchi et al., 2021; Jiang & Zhang, 2020; Chen et al., 2020b; Rossi et al., 2021; Um et al., 2023).

Despite the significant advancements in model design for GNNs in MDI, the generalizability of these GNN-based techniques across varied domains, especially those frequently encountering real-world missing data, bio-medical domain, remains under-explored. We undertook a comparative analysis between the widely-researched citation network domain and the bio-medical domain, where generalizability is paramount. As illustrated in Figure 1, within the citation network where a graph structure is readily available, the performance of both the pioneering Gaussian Mixture Model-based GCNMF (Taguchi et al., 2021) and the recently proposed Dirichlet energy-minimizing Feature Propagation (FP) (Rossi et al., 2021) demonstrates remarkable resilience across an extensive spectrum of missing feature rates (spanning 0.1 to 0.9). Notably, they outperform the gold standard tabular-based imputation technique Mean (Little & Rubin, 2019) and the popular Generative Adversarial Network-based approach, GAIN (Yoon et al., 2018). The efficacy of these graph-based techniques

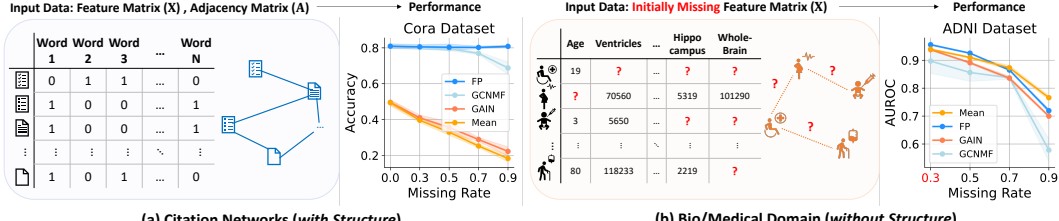

Figure 1: Comparison of citation network and bio-medical domain regarding their input data and performance. The Cora dataset focuses on node classification, while the Alzheimer's Disease Neuroimaging Initiative (ADNI) dataset emphasizes patient diagnosis classification. Unlike the Cora dataset, the ADNI dataset has an initial missing feature rate of 30%.

stems from their ability to leverage the graph structure, incorporating a message-passing scheme that allows similar nodes to inform their imputation, leading to a smoother representation. In essence, when a suitable graph structure is available, the adoption of graph-based imputation techniques becomes imperative.

In contrast, the bio-medical domain presents unique challenges. Frequently encountering missing data scenarios—often seen in patient information datasets (Wiens, 2003; Cismondi et al., 2013b)—the majority of the available data is tabular. This tabular format, rife with initial missing values and devoid of any inherent graph structure, induces graph-based imputation works to struggle to achieve the same efficacy they exhibit in aforementioned domains like citation networks. Evidence from Figure 1 (b) underscores this distinction: the performance disparity between imputation methods utilizing graphs[1] (e.g., GCNMF, FP) and methods without relying on graphs (e.g., Mean, GAIN) is marginal. Alarmingly, we note that, under severe missing rates (e.g., 0.9), graph-based techniques like GCNMF and FP exhibit performance degradation, even underperforming the rudimentary Mean baseline. This limitation primarily stems from the lack of a graph structure relevant to downstream tasks, a staple in previously successful applications of graph-based imputation. These observations spotlight that modern graph-based imputation techniques are yet to achieve generalization in the bio-medical realm, where missing data challenges are ubiquitous. It naturally posits the central question: ***Is it feasible to craft a more insightful feature matrix and associated graph structure, thereby leveraging the potential of graph-based imputation techniques in the bio-medical domain?***

Here, we introduce GRASS, a novel approach that offers an orthogonal way to leverage and generalize existing graph-based imputation methods to real-world missing scenarios—such as those in the bio-medical domain—where both initially missing features and graph structure are prevalent. Instead of directly constructing a graph structure based on the current incomplete features, which would be suboptimal, we commence with training on the tabular data using Multi-Layer Perceptron (MLP) layers. During this training process, a valuable task-relevant byproduct that naturally emerges is *gradient information with respect to the features*. This gradient information serves as a direct indicator of how variations in features impact the prediction of the downstream tasks at hand. We utilize this gradient information by concatenating it to the original feature matrix, thereby creating a feature perspective graph. Subsequently, we apply column-wise feature propagation to produce a warmed-up feature matrix. At this moment, we create a similarity-based adjacency matrix of samples derived from this updated feature matrix. Now, equipped with this warmed-up feature matrix and the new graph structure, we stand ready to harness the potential of cutting-edge graph-based imputation techniques, extending our reach to real-world missing data scenarios.

In summary, our contributions are three-fold:

- We, for the first time, explore the generalizability of recent graph-based imputation models in the context of real-world bio-medical tabular data with missing values.

- We propose a novel approach for constructing a graph structure that does not solely depend on the initially incomplete feature matrix by utilizing feature gradient information.

- We demonstrate that GRASS can serve as an effective initial starting point in a model-agnostic fashion, thereby enhancing performance in downstream tasks across multiple bio-medical datasets.

---

[1]We generate a similarity-based kNN graph due to the absence of given graphs.

## 2 RELATED WORKS

**Tabular-based Data Imputation.** The challenge of missing data imputation has a long history and many early approaches for tabular data are rooted in statistical methods (Efron, 1994; Little & Rubin, 2019). These methods often leverage the distribution of non-missing values to impute missing ones. Recent machine learning-based imputation techniques include kNN-based approaches (Troyanskaya et al., 2001; Keerin et al., 2012; Malarvizhi & Thanamani, 2012), GAIN (Yoon et al., 2018), which employs Generative Adversarial Networks (Goodfellow et al., 2020), and MIWAE (Mattei & Frellsen, 2019), which utilizes a Deep Latent Variable model (Kingma & Welling, 2013). There have also been efforts to adapt graph structures to tabular data for imputation; for example, GRAPE (You et al., 2020) introduces a bipartite graph connecting samples and features, while the more recent IGRM (Zhong et al., 2023) extends GRAPE by adding a friend network to capture relationships between samples. However, as these methods heavily rely on the input feature matrix as a main resource, in cases where a significant proportion of data is missing, the imputation quality tends to degrade, negatively affecting downstream tasks' performance. Notably, compared to the graph domain, most of these studies emphasize either imputation or regression. This is because the task of imputing continuous values closely aligns with regression, simplifying both training and evaluation. However, another pivotal downstream task, *i.e.*, classification, remains underexplored in the realm of tabular data with missing features.

**Graph-based Data Imputation.** From the viewpoint of graph-based imputation, GCNMF (Taguchi et al., 2021) tackles missing features by assuming a Gaussian distribution for each feature channel while aligning it with Graph Convolutional Networks (GCN) (Kipf & Welling, 2016a). PaGNN (Jiang & Zhang, 2020) proposes a partial aggregation scheme derived from neighborhood reconstruction. FP (Rossi et al., 2021) iteratively diffuses known features to unknown features, followed by GNN layers. Recently, PCFI (Um et al., 2023) builds upon FP to introduce channel-wise diffusion confidence to handle scenarios with higher missing feature rates. However, channel-wise diffusion operates on fully connected graphs, potentially incorporating irrelevant or noisy information between channels. Additionally, they carry a strong inductive bias toward readily available graph structures, limiting their generalizability. As mentioned above, the application domain of these works primarily focuses on Citation (Sen et al., 2008) and Co-Purchase networks (Shchur et al., 2018) where features are text-based, a situation less reflective of realistic cases where features are initially missing.

**Bio-medical Data Imputation.** In the medical domain, several research efforts have been made to address missing data. Multiple imputation techniques are suggested by (Janssen et al., 2010), while (Cismondi et al., 2013a) employs statistical approaches for imputation. The MICE algorithm (Van Buuren & Groothuis-Oudshoorn, 2011) is also widely applied in this context. On the biology side, a prominent issue related to missing data is the occurrence of dropout events in single-cell RNA-sequencing datasets, where zero values are often falsely recorded as missing. Among the various methods proposed (Li & Li, 2018; van Dijk et al., 2018; Wang et al., 2021; Yun et al., 2023), scGNN (Wang et al., 2021) and scFP (Yun et al., 2023) employ Graph Auto-Encoders (GAE)(Kipf & Welling, 2016b) and FP(Rossi et al., 2021) to impute these false zeros. Despite these efforts, the use of graph-based data imputation techniques remains underexplored. This is largely due to the absence of a network structure and a reliance on input feature matrices. Such limitations widen the gap between recent advances in graph-based imputation and real-world applications where data is often missing.

## 3 METHODOLOGY

In this section, we introduce GRASS, a novel algorithm designed to bridge the gap between recent graph-based imputation methods and real-world missing data scenarios. Initially, we employ a Multi-Layer Perceptron (MLP) to extract the feature gradient, a crucial supplement for graph structure used for imputation (Sec 3.1). Subsequently, we implement a Column-wise Feature Propagation grounded on the gradient-informed graph (Sec 3.2). Then, we derive a warmed-up feature matrix alongside with a kNN graph, setting the stage for seamless integration with contemporary graph-based imputation techniques (Sec 3.3). The comprehensive framework of GRASS is depicted in Figure 3.

**Task: Classification with Tabular Data Containing Initial Missing Features.** Given an initially missing feature matrix $\mathbf{X} \in \mathbb{R}^{N \times F}$, where $N$ denotes the total samples and $F$ the feature dimensions,

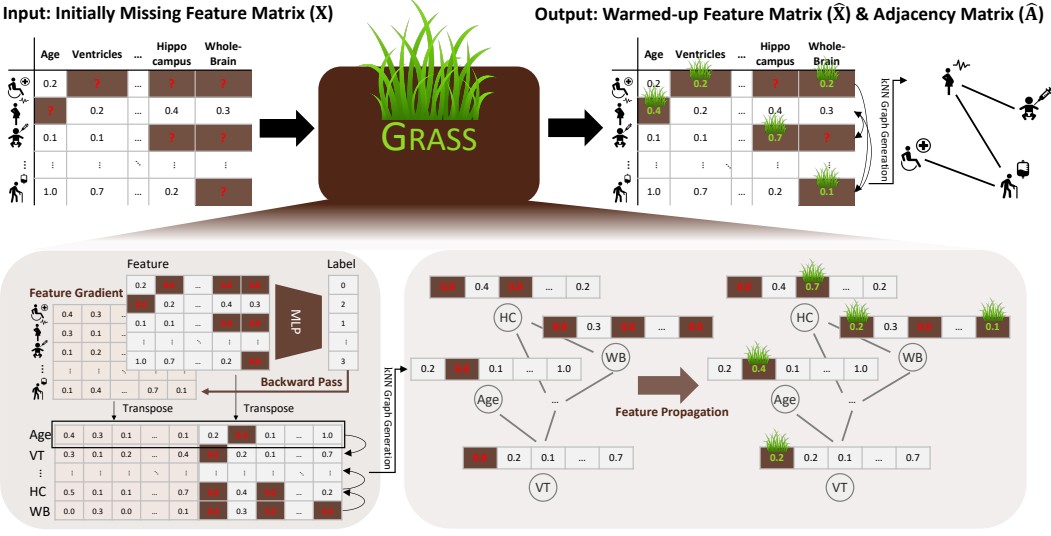

Figure 2: Overall framework of GRASS. Given an initially missing feature matrix, we first train a simple MLP to obtain the feature gradient. By concatenating these gradients with the initial matrix, we create a graph from a feature-wise perspective. After employing Column-wise Feature Propagation, we obtain a warmed-up feature matrix and an adjacency matrix. These serve as the foundational feature and adjacency matrices for current graph-based imputation methods.

the goal of GRASS is to produce a warmed-up feature matrix accompanied by a sample-wise graph structure. This enhanced matrix corresponding graph structure enables existing graph-based imputation methods to seamlessly utilize them as an initial reference point.

## 3.1 FEATURE GRADIENT AS A SUPPLEMENT

While facing the challenge of *initially missing* feature matrix, the direct utilization of this matrix for downstream tasks can lead to suboptimal results. Prior imputation is imperative. Naturally, one might consider the latest graph-based imputation techniques (Taguchi et al., 2021; Jiang & Zhang, 2020; Rossi et al., 2021; Um et al., 2023) given their prowess in receiving messages (or representations) from neighboring samples. However, as depicted in Figure 1, constructing a graph based only on given features becomes increasingly challenging with high feature missing rates, at times even counteracting the imputation process. While it might be tempting to first address missing values simply by employing Mean (Little & Rubin, 2019), kNN (Troyanskaya et al., 2001), or other statistical methods (Van Buuren & Groothuis-Oudshoorn, 2011; White et al., 2011) prior to constructing a graph, this approach remains constrained by the use of incomplete features. Furthermore, it often lacks task-specific insights, which is particularly crucial when targeting downstream tasks like classification. In light of this, we identify *feature gradient*, obtained during backpropagation, as a pivotal, task-aligned resource. These gradients indicate how subtle shifts in features impact the model's predictions, highlighting the salience of individual features in loss minimization. By leveraging these gradients, we can devise a graph structure that encapsulates not just observed feature information from an initial state but also the feature saliency in relation to our targeted downstream task. Consequently, we begin with a formal definition of a feature gradient.

**Definition 3.1.** *The feature gradient, denoted as $\nabla_{\mathbf{X}}$, represents the partial derivatives of the loss function concerning each feature in the input matrix and is mathematically defined as $\nabla_{\mathbf{X}} = \frac{\partial \mathcal{L}}{\partial \mathbf{X}}$.*

Building upon **Definition** 3.1, we derive feature gradient through the training of a straightforward MLP [2]. During this training process, the ensuing proposition emerges:

---

[2]While training the MLP layers, we utilized zero imputation for the initially missing values, capitalizing on its computational efficiency and steering clear of presumptions associated with missing completely at random (MCAR).

**Proposition 1.** *Consider a 2-layer Multi-Layer Perceptron (MLP). The output for each layer is formulated as:* $\mathbf{Z}' = \sigma(\mathbf{X}\mathbf{W}' + \mathbf{b}'), \mathbf{Z}'' = \mathbf{Z}'\mathbf{W}'' + \mathbf{b}''$ *where the trainable weight matrices are denoted as* $\mathbf{W}' \in \mathbb{R}^{F \times D}$ *and* $\mathbf{W}'' \in \mathbb{R}^{D \times C}$, *and bias vectors are represented by* $\mathbf{b}' \in \mathbb{R}^D$ *and* $\mathbf{b}_2 \in \mathbb{R}^C$. *The activation function,* $\sigma$, *is chosen as the ReLU function,* F is the feature dimension, *and* $D$ *specifies the dimension. Upon applying the softmax function, we derive the prediction probability matrix* $\hat{\mathbf{Y}} \in \mathbb{R}^{N \times C}$, *with* $C$ *indicating the number of classes.* $\mathbf{Y} \in \mathbb{R}^{\mathbf{N} \times \mathbf{C}}$ is a label matrix. *Using cross-entropy as the loss function, the feature gradient, represented as* $\boldsymbol{\nabla}_{\mathbf{X}} \in \mathbb{R}^{N \times F}$, *can be computed as:*

$$\boldsymbol{\nabla}_{\mathbf{X}} = ((\hat{\mathbf{Y}} - \mathbf{Y}) \cdot \mathbf{W}''^{\top}) \odot (\mathbf{X}\mathbf{W}' + \mathbf{b}' > 0) \cdot \mathbf{W}'^{\top}$$

Please refer to Appendix A.1 for the detailed proof.

A central observation from **Proposition**1 is the dynamic nature of the feature gradient matrix across MLP training epochs, despite the static nature of the initially provided missing feature matrix $\mathbf{X}$. This dynamic is attributed to adjustments in trainable weight parameters (e.g., $\mathbf{W}_1, \mathbf{W}_2$), which in turn influence feature gradient variations. Although the imparted information undergoes change every epoch, persistently storing these gradients across epochs incurs substantial memory overhead, $\mathcal{O}(NF)$. Moreover, there's no guarantee of consistent gradient quality improvement with each epoch. To address this, we selectively store feature gradient[3] only when the MLP's performance on predicting the validation set improves, leveraging them as pivotal cues to enhance downstream task efficacy. In essence, after training MLP, we consolidate the stacked feature gradient, averaging them to yield $\overline{\boldsymbol{\nabla}}_{\mathbf{X}} \in \mathbb{R}^{N \times F}$, a matrix accordant in shape with the original feature matrix. This then acts as an adjunct to the initially missing feature matrix, as elucidated in the following section.

## 3.2 COLUMN-WISE FEATURE PROPAGATION

In the context of classification, post-imputation matrices often undergo message-passing, especially when adopting GNNs as classifiers. While GNNs typically employ adjacency matrices constructed from a node's perspective, they might neglect significant inter-feature relationships. For example, in the ADNI dataset (Petersen et al., 2010), the relationship between attributes like 'Age' and 'Ventricles' has been documented to indicate potential brain volume loss as one ages (Nestor et al., 2008; Bjork et al., 2003). To capture such crucial inter-feature dynamics, we introduce a column-wise graph structure. Here, given that the initial columns (i.e., features) have missing values, we address this challenge by a supplement, the feature gradient we derived earlier, as follows:

$$\mathbf{A}^{feat} = k_{\text{col}}\text{-nearest-neighbor}(\overline{\boldsymbol{\nabla}}_{\mathbf{X}}^{\top} \| \mathbf{X}^{\top}) \tag{1}$$

where $k_{\text{col}}$-nearest-neighbor($\cdot$) denotes the connection of $k_{\text{col}}$ neighbors for each feature channel, established using cosine similarity, with $k_{\text{col}}$ as a hyperparameter. Given the feature-wise graph, we employ FP (Rossi et al., 2021) to estimate missing features across iterations by capturing inter-feature relationships in a *column-wise fashion* while preserving known values, which is depicted as follows:

$$\mathbf{X}^{(i+1)\top} = \tilde{\mathbf{A}}^{feat}\mathbf{X}^{(i)\top},$$
$$\mathbf{X}^{(i+1)\top}_{v,d} = \mathbf{X}^{(0)\top}_{v,d}, \forall v \in \mathcal{V}_{known,d}, \forall d \leq F \tag{2}$$

where $\tilde{\mathbf{A}}^{feat} = \mathbf{D}^{-1/2}\mathbf{A}^{feat}\mathbf{D}^{-1/2} \in \mathbb{R}^{F \times F}$ is symmetrically normalized weighted adjacency, having cosine similarity as a weight, with a self-loop with added degree matrix $\mathbf{D}$. At iteration $i$, the matrix is represented as $\mathbf{X}^{(i)\top} \in \mathbb{R}^{F \times N}$. The set $\mathcal{V}_{known,d}$ contains nodes with known feature values for the $d$-th channel. After $K$ iterations and another transposition, we obtain the imputed output $\hat{\mathbf{X}} = \mathbf{X}^{(K)} \in \mathbb{R}^{N \times F}$, which we term the *warmed-up matrix*. Given our approach uses a custom $k$NN graph, discussions about convergence can be found in Appendix A.2.

---

[3]L2-normalization was applied during feature gradient storage to maintain consistent feature scales and retain the original vector's directionality.

## 3.3 WARMED-UP FEATURE MATRIX AND ADJACENCY MATRIX

Real-world tabular datasets often contain a blend of numerical (e.g., Age, Blood pressure) and categorical features (e.g., Gender, Blood type). To address the variability, we introduce a clamping method tailored for categorical features that aligns with our Column-wise Feature Propagation approach. Given iterative multiplications with the normalized adjacency matrix, one-hot encoded categorical columns can yield continuous imputed values. In the bio-medical field, consideration of such continuous value-based imputation can be vital, especially when examining samples from different tissues or cell types. For instance, if a tissue type is labeled '0' for "epithelial" and '1' for "mesenchymal", an imputed value of 0.5 would be biologically meaningless and not represent any known tissue type. Thus, to preserve the original scale of these features, for a categorical index $c$ which we obtain during the preprocessing of numerical and categorical mixed type tabular data, with corresponding bin count $c_b$ for the original column, the predicted probability vector for a sample $j$ from the continuous imputed matrix is given by: $\tilde{\mathbf{x}}_c = \text{softmax}(\hat{\mathbf{X}}_{j,c:c+c_b}) \in \mathbb{R}^{c_b}$ Subsequently, the clamping process is as below:

$$\hat{\mathbf{X}}_{j,c:c+c_b} = \begin{cases} \text{OneHot}(\text{argmax}(\tilde{\mathbf{x}}_c)), & \text{if } \max(\tilde{\mathbf{x}}_c) \geq \theta \\ \underbrace{[?, \ldots, ?]}_{c_b \text{ times}}, & \text{otherwise} \end{cases} \tag{3}$$

where OneHot($\cdot$) function represents one-hot encoding based on a threshold, $\theta$. The symbol ? indicates retained initial missing values, emphasizing our aim to preserve inherent uncertainties. Using this method, we aptly handle categorical columns, ensuring imputations align with their original format. After obtaining the clamped matrix $\hat{\mathbf{X}}$, we create a row-wise (i.e., sample-wise) graph structure, deriving edge indices to be aligned with samples using the imputed warmed-up matrix as a resource. This serves as the foundational graph for contemporary graph-based imputation methods, which can be described as:

$$\hat{\mathbf{A}} = k_{\text{row}}\text{-nearest-neighbor}(\hat{\mathbf{X}}),$$

where $k_{\text{row}}$-nearest-neighbor($\cdot$) denotes the connection of $k_{\text{col}}$ neighbors for each sample using cosine similarity, with $k_{\text{row}}$ as a hyperparameter. With the clamped matrix and sample-wise graph structure, we offer a refined starting point for current graph-based imputation methods. It is essential to recognize that our approach still provides room for contemporary graph-based imputation methods for their own imputations, as shown in Figure 3. Column-wise FP might not always impute all missing values, especially when nodes are not connected to the nodes containing known values. Additionally, if the clamping process doesn't meet a set threshold (Equation 3), this situation would occur. Our goal is to enhance the potential of current graph-based imputation techniques. By introducing a warmed-up matrix ($\hat{\mathbf{X}}$) using feature gradient and a related graph structure ($\hat{\mathbf{A}}$), we strengthen the performance of current imputation strategies in classification tasks. We present the detailed algorithm for the overall process of GRASS in Appendix A.4.

## 4 EXPERIMENTS

### 4.1 EXPERIMENTAL SETTINGS

**Datasets.** We evaluate GRASS on nine datasets in total containing initial missing data. Four bio datasets are from the single-cell RNA-seq domain: Mouse ES (Klein et al., 2015), Pancreas (Luecken et al., 2022), Baron Human (Baron et al., 2016), and Mouse Bladder (Han et al., 2018). Five of these are from the medical domain: Hepatitis from the UCI Machine Learning repository, related to liver diseases (hep, 1988). Alzheimer's Disease Neuroimaging Initiative (ADNI) related to brain diseases (Petersen et al., 2010). Autism Brain Imaging Data Exchange (ABIDE) is also related to brain diseases (Di Martino et al., 2014; 2017). Dynamic contrast-enhanced magnetic resonance images of breast cancer patients with tumor locations (Duke Breast) (Saha et al., 2018; 2021). Breast Cancer from the UCI repository, related to breast diseases (Zwitter & Soklic, 1988). For the dataset split, we randomly generated 5 different splits with train/val/test ratio in 10%, 10%, 80% [4]. Statistics for each dataset and details are provided in Appendix A.3.

---

[4]It is noteworthy that while the scRNA-seq domain has traditionally been unsupervised, the increasing availability of public scRNA-seq datasets has shifted the trend towards supervised machine learning models in recent research (Cao et al., 2022).

**Compared Methods.** To verify whether our algorithm enhances current graph-based imputation methods, we compare it with established baselines such as Label Propagation (LP) (Zhu, 2005), GCNMF (Taguchi et al., 2021), PaGNN (Jiang & Zhang, 2020), Neighborhood Mean (NM), Zero Imputation with GCN layers (Zero) (Rossi et al., 2021), FP (Rossi et al., 2021), and PCFI (Um et al., 2023). Given our focus on tabular data, we also include common methods like Mean (Little & Rubin, 2019), kNN (Troyanskaya et al., 2001), GAIN (Yoon et al., 2018), MIWAE (Mattei & Frellsen, 2019), and bipartite graph-based approaches like GRAPE (You et al., 2020) and its recent enhanced version, IGRM (Zhong et al., 2023). To align with our focus on downstream tasks, we appended a logistic classifier to methods that exclusively target imputation.

**Hyper-parameters.** For graph-based imputation methods, we generated a $k$NN graph, selecting $k$ from 1, 3, 5, 10. We set a consistent dropout rate of 0.5 and a dimension of 64 across all methods. While other hyperparameters were tuned based on the original paper's recommendations, for our model, we similarly explored values for $k_{\text{col}}$ and $k_{\text{row}}$ within 1, 3, 5, 10. The clamping process's threshold, $\theta$, was tested among 0.0, 0.2, 0.4, 0.6, 0.8. We set the number of iterations, $K$, to 40, as advised in the FP paper. Optimal hyperparameters for the best and most improved models are detailed in Appendix A.3.

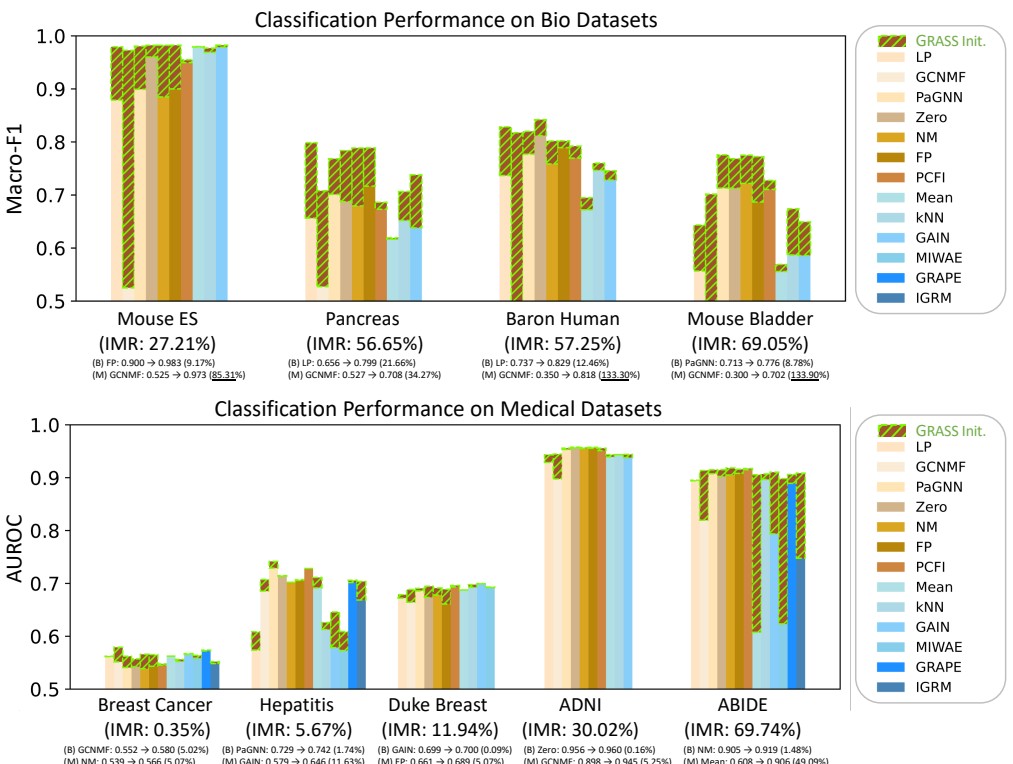

Figure 3: Classification performance on Medical/Bio dataset with initially missing rate (IMR). (B) denotes the "Best" performing baseline while (M) denotes the "Most Improved" baseline with their relative improvement percentage. A percentage is underlined if it surpasses 80%.

## 4.2 CLASSIFICATION PERFORMANCE

As depicted in Figure 3, graph-based imputation methods (e.g., FP, PCFI) naturally excel in classification tasks compared to tabular-based models, largely due to their message-passing mechanisms. The bio domain, which only comprises numerical data, showed notable performance improvement compared to the mixed-type medical domain. Yet, in certain datasets like Mouse ES, advanced graph-based imputations like PaGNN and FP occasionally do not surpass basic methods tailored to tabular data, a trend also noticed in Figure 1. However, when integrated with the warmed-up matrix and adjacency matrix from GRASS, their performance significantly improves, even surpassing tabular-based models. It's noteworthy that while recent methods like FP and PCFI outperform GCNMF in citation networks with up to a 99% missing rate, GCNMF's potential remains significant

in the bio-medical domain. The appropriate graph structure, combined with the Gaussian Mixture Model, can significantly enhance performance, leading to a remarkable 133.90% improvement in the Mouse Bladder dataset. Although our model was primarily developed to refine and offer a more tailored adjacency matrix, tabular-based methods also benefit from using a warmed-up matrix as their starting point, as seen in the ABIDE dataset. In summary, while the adaptability of current graph-based imputation models hasn't been thoroughly explored, and they occasionally underperform simpler tabular methods, incorporating GRASS from the beginning can maximize their capabilities.

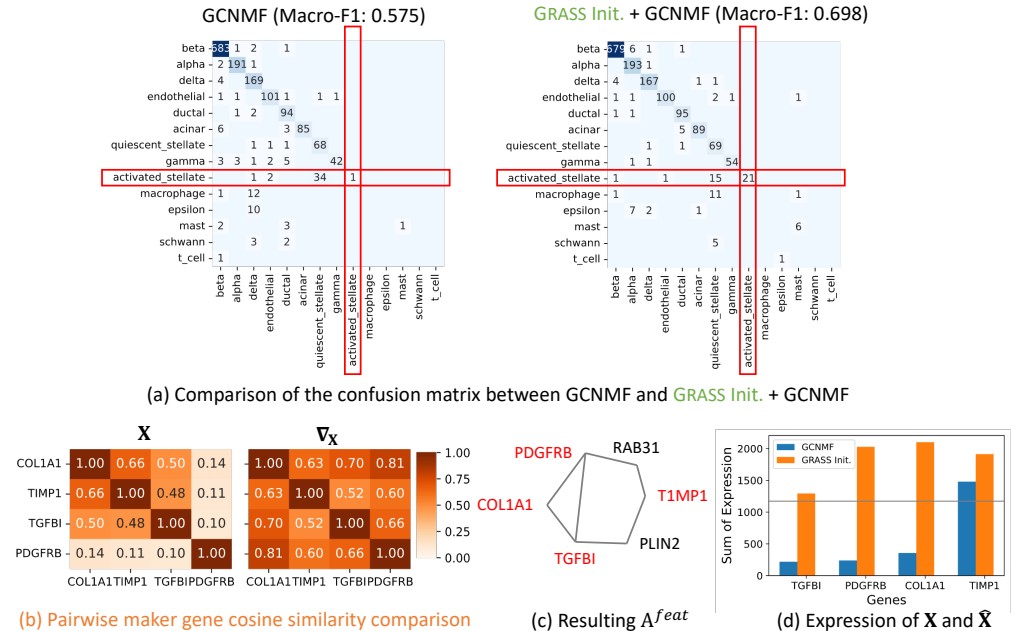

(a) Comparison of the confusion matrix between GCNMF and GRASS Init. + GCNMF

(b) Pairwise maker gene cosine similarity comparison

(c) Resulting $A^{feat}$

(d) Expression of $\mathbf{X}$ and $\hat{\mathbf{X}}$

Figure 4: Exploring the influence of feature gradient. (a) Confusion matrix comparison between original GCNMF and its GRASS initialized version, illustrating the latter capturing more rare cell-type. (b) Pairwise marker gene cosine similarity comparison between original feature matrix ($\mathbf{X}$ and feature gradient($\nabla_{\mathbf{X}}$), resource for the column-wise graph. (c) Resulting $\mathbf{A}^{feat}$ via utilizing feature gradient as a supplement. (d) Expression of four marker genes being amplified after column-wise Feature Propagation. All experiments were conducted on the Pancreas dataset. Marker genes, which are key factors for classifying "activated stellate" cell type, were identified based on existing research linking these genes to the activated hepatic stellate cell (HSC).

## 4.3   WHY FEATURE GRADIENT MATTERS?

We investigate the contributions of *feature gradients* in addressing the missing feature problem. We first analyze the underlying information of feature gradients with class (cell-type) representations in Figure 5. The class representations of feature gradients (Figure 5 (b)) are more discriminative than the original input matrix (Figure 5 (a)). Then, with the use of feature gradients, the final warmed-up matrix (Figure 5 (c)) can further enhance the intra-class distinction. The observations indicate that the feature gradients are beneficial for learning more distinct class representations than the original input features. It is worth noting that we can extract the feature gradients based solely on the input matrix without any external information, demonstrating the broad application of the proposed model.

We delved deeper into the predictions made by GRASS to elucidate the impact of feature gradients. Figure 4 (a) showcases confusion matrices comparing GCNMF, the model showing the most improvement, with our warmed-up matrix. A significant factor in this enhanced performance is the better recognition of rare cell types like 'activated stellate'. Figure 4 (b) illustrates that expression of marker genes for 'activated stellate' such as COL1A1, TIMP1, TGFBI, and PDGFRB exhibit higher cosine similarity in feature gradient compared to its original matrix. This was achievable owing to its ability to incorporate task-relevant information, i.e., 'activated stellate' cell-type information, which is brought from the label supervision. By leveraging this task-relevant gradient information, in Figure 4 (c), direct connections (i.e., 1-hop neighbors) have been formed among three marker

genes, while one displays 2-hop relationships. Consequently, after feature propagation iterations, these genes exhibit increased expression levels owing to neighborhood aggregation. This crucially pinpoints the 'activated stellate' cell type, as demonstrated in Figure 4 (d), surpassing the average expression value across all genes (gray line). Table 1 underscores the importance of each module in terms of AUROC, showing the effectiveness of column-wise feature propagation. Utilizing feature gradients with clamping techniques proves especially beneficial in bio-medical domains.

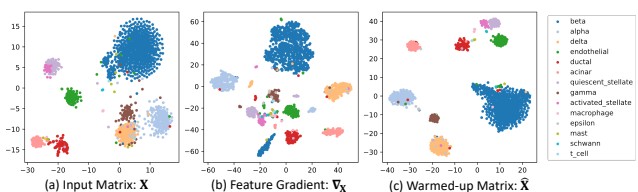

Figure 5: t-SNE of cell-type representation in Pancreas data with input matrix, feature gradient, and warmed-up matrix.

Table 1: Ablation study of GRASS. Two best-performing models, GCNMF and PaGNN, are used for the backbone model. (w/o room) denotes the remaining missing values are imputed as zeros.

| Model Variants | Breast Cancer | Hepatitis |
|---|---|---|
| Row only | 0.500+0.00 | 0.667+0.10 |
| Col only | 0.524+0.08 | 0.603+0.20 |
| Col+$\nabla_{\mathbf{X}}$ | 0.540+0.10 | 0.627+0.11 |
| Col+$\nabla_{\mathbf{X}}$+Clamp (w/o room) | 0.577+0.05 | 0.736+0.07 |
| Col+$\nabla_{\mathbf{X}}$+Clamp (w/ room) | 0.579+0.08 | 0.742+0.058 |

### 4.4 SENSITIVITY ANALYSIS

Figure 6 (a) illustrates the sensitivity of hyperparameters $k_{\mathrm{col}}$ and $k_{\mathrm{row}}$, which are responsible for the generation of neighboring edges for column-wise (feature-wise) and row-wise (sample-wise) graphs, respectively. Our observations suggest that GRASS exhibits relative robustness to variations in the number of neighbors within the recommended range $\{1, 3, 5, 10\}$. Nonetheless, in situations with extensive missing data, such as the ABIDE dataset, which has an initial missing rate of 69.74%, opting for a larger $k$ for column-wise graphs might be inadvisable. This is because the resulting connected components may be more uncertain, heavily populated with missing data, and potentially prone to the over-smoothing issue. The implications of selecting a higher $k$ value, especially concerning convergence and over-smoothing, are further discussed in Appendix A.2. Meanwhile, Figure 6 (b) emphasizes the significance of selecting an appropriate clamping threshold. A larger $\theta$ tends to retain more uncertainties, preserving more original missing values. Conversely, a smaller $\theta$ might prompt premature imputation by GRASS, possibly limiting the subsequent graph-based imputation methods' ability to further refine using the warmed-up matrix and adjacency matrix offered by GRASS, as showcased in Table 1.

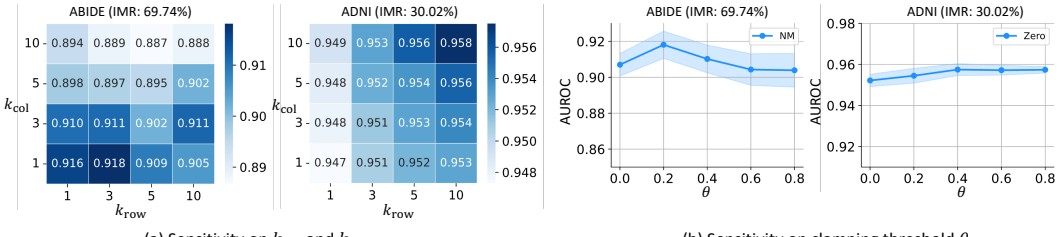

(a) Sensitivity on $k_{\mathrm{col}}$ and $k_{\mathrm{row}}$      (b) Sensitivity on clamping threshold $\theta$

Figure 6: Sensitivity analysis on hyperparameters used in GRASS. AUROC is measured in both datasets.

## 5 CONCLUSION

Graph-based imputation approaches have gained increasing popularity to fill in the missing feature by aggregating the information from its neighborhood nodes. However, most real-world scenarios do not satisfy their pre-assumption of a readily available graph structure, which limits the partial usage of current methods. To tackle the bottleneck, we propose an innovative GRASS algorithm to generalize graph-based imputation to realistic cases like medical and bio-domain datasets. GRASS begins with tabular data and leverages their feature gradients to construct graph structures from both feature- and sample-oriented perspectives. Extensive empirical investigations consistently validate the effectiveness of our proposal. We believe GRASS can serve as an attractive starting point for future graph-based imputation works.

**Ethical Statement.** In accordance with the ICLR Code of Ethics, we confirm that our research adheres to its prescribed guidelines. While our proposed algorithm can recover missing features, particularly in the bio-medical domain, we advise not using it for actions that might lead to negative societal consequences, such as the unauthorized sharing of private information.

**Reproducibility Statement.** For clarity and reproducibility, we've detailed the three proposed modules in Sections 3.1, 3.2, and 3.3. A theoretical proof for deriving the feature gradient can be found in Appendix A.1, while the complete pseudocode is available in Appendix A.4. Information regarding the experimental settings is laid out in Appendix A.3. The code can be accessed at `https://anonymous.4open.science/r/grass-iclr-41D5`.

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

# A APPENDIX

## A.1 PROOF OF PROPOSITION 1

**Proposition 1.** *Consider a 2-layer Multi-Layer Perceptron (MLP). The output for each layer is formulated as:* $\mathbf{Z}' = \sigma(\mathbf{X}\mathbf{W}' + \mathbf{b}'), \mathbf{Z}'' = \mathbf{Z}'\mathbf{W}'' + \mathbf{b}''$ *where the trainable weight matrices are denoted as* $\mathbf{W}' \in \mathbb{R}^{F \times D}$ *and* $\mathbf{W}'' \in \mathbb{R}^{D \times C}$, *and bias vectors are represented by* $\mathbf{b}' \in \mathbb{R}^D$ *and* $\mathbf{b}_2 \in \mathbb{R}^C$. *The activation function,* $\sigma$, *is chosen as the ReLU function, and* $D$ *specifies the dimension. Upon applying the softmax function, we derive the prediction probability matrix* $\hat{\mathbf{Y}} \in \mathbb{R}^{N \times C}$, *with* $C$ *indicating the number of classes. Using cross-entropy as the loss function, the feature gradient, represented as* $\nabla_{\mathbf{X}} \in \mathbb{R}^{N \times F}$, *can be computed as:*

$$\nabla_{\mathbf{X}} = ((\hat{\mathbf{Y}} - \mathbf{Y}) \cdot \mathbf{W}''^{\top}) \odot (\mathbf{X}\mathbf{W}' + \mathbf{b}' > 0) \cdot \mathbf{W}'^{\top}$$

*Proof.* Given a row-vector, $\mathbf{x} \in \mathbb{R}^{1 \times F}$, consider the following application of the chain rule:

$$\frac{\partial \mathcal{L}}{\partial \mathbf{x}} = \frac{\partial \mathcal{L}}{\partial \mathbf{z}''} \cdot \frac{\partial \mathbf{z}''}{\partial \mathbf{z}'} \cdot \frac{\partial \mathbf{z}'}{\partial \mathbf{x}}$$

To compute $\frac{\partial \mathcal{L}}{\partial \mathbf{z}''}$, let's begin by considering a specific class index $n$, when $n$ ranges from 1 to $C$, the total number of classes.

$$\begin{aligned}
\frac{\partial \mathcal{L}}{\partial z_n''} &= \frac{\partial \mathcal{L}}{\partial \hat{y}_n} \cdot \frac{\hat{y}_n}{\partial z_n''} \\
&= -\sum_{i=1}^{C} y_i * \frac{\partial \log(\hat{y}_i)}{\partial \hat{y}_i} * \frac{\partial \hat{y}_i}{\partial z_n''} = -\sum_{i=1}^{C} \frac{y_i}{\hat{y}_i} * \frac{\partial \hat{y}_i}{\partial z_n''}
\end{aligned}$$

To determine $\frac{\partial \hat{y}_i}{\partial z_n''}$, the gradient with respect to the softmax function for each class $i$ in total $C$ classes can be computed:

I. When $i = n$,

$$\begin{aligned}
\frac{\partial \hat{y}_i}{\partial z_i''} &= \frac{\partial}{\partial z_i''} \left( \frac{e^{z_i''}}{\sum_{j=1}^{C} e^{z_j''}} \right) \\
&= \frac{e^{z_i''} * \sum_{j=1}^{C} e^{z_j''} - \left( e^{z_i''} \right)^2}{\left( \sum_{j=1}^{C} e^{z_j''} \right)^2} \\
&= \frac{e^{z_i''}}{\sum_{j=1}^{C} e^{z_j''}} * \frac{\sum_{j=1}^{C} e^{z_j''} - e^{z_i''}}{\sum_{j=1}^{C} e^{z_j''}} \\
&= \hat{y}_i * (1 - \hat{y}_i)
\end{aligned}$$

II. When $i \neq n$,

$$\frac{\partial \hat{y}_i}{\partial z_n''} = \frac{0 * \sum_{j=1}^{C} e^{z_j''} - e^{z_i''} * e^{z_n''}}{\left(\sum_{j=1}^{C} e^{z_j''}\right)^2}$$

$$= -\frac{e^{z_i''} * e^{z_n''}}{\left(\sum_{j=1}^{C} e^{z_j''}\right)^2}$$

$$= -\frac{e^{z_i''}}{\sum_{j=1}^{C} e^{z_j''}} * \frac{e^{z_n''}}{\sum_{j=1}^{C} e^{z_j''}}$$

$$= -\hat{y}_i * \hat{y}_n$$

We can subsequently consolidate two separate cases as follows:

$$\frac{\partial L}{\partial z_n''} = -\sum_{i=1}^{C} \frac{y_i}{\hat{y}_i} * \frac{\partial \hat{y}_i}{\partial z_n''}$$

$$= -\frac{y_n}{\hat{y}_n} * \hat{y}_n * (1 - \hat{y}_n) + \sum_{i \neq n}^{c} \frac{y_i}{\hat{y}_i} * \hat{y}_i * \hat{y}_n$$

$$= -y_n + y_n * \hat{y}_n + \sum_{i \neq n}^{c} y_i * \hat{y}_n$$

$$= -y_n + \sum_{i=1}^{C} y_i * \hat{y}_n$$

$$= \hat{y}_n - y_n$$

The vector form for the same is:

$$\frac{\partial \hat{\mathbf{y}}}{\partial \mathbf{z}''} = \hat{\mathbf{y}} - \mathbf{y}$$

Now, the gradient with respect to the output of the hidden layer, $\frac{\partial \mathbf{z}''}{\partial \mathbf{z}'}$ is directly given by:

$$\frac{\partial \mathbf{z}''}{\partial \mathbf{z}'} = \mathbf{W}''^\top$$

Lastly, to obtain $\frac{\partial \mathbf{z}'}{\partial \mathbf{x}}$, we need to consider the ReLU activation in the hidden layer:

$$\frac{\partial \mathbf{z}'}{\partial \mathbf{x}} = \frac{\partial \mathbf{z}'}{\partial \sigma(\mathbf{x}\mathbf{W}' + \mathbf{b}')} \cdot \frac{\partial \sigma(\mathbf{x}\mathbf{W}' + \mathbf{b}')}{\partial \mathbf{x}} = (\mathbf{x}\mathbf{W}' + \mathbf{b}' > 0) \cdot \mathbf{W}'^\top$$

Combining these results yields the feature gradient in row-vector ($\mathbb{R}^{1 \times F}$) format:

$$\frac{\partial \mathcal{L}}{\partial \mathbf{x}} = \frac{\partial \mathcal{L}}{\partial \mathbf{z}''} \cdot \frac{\partial \mathbf{z}''}{\partial \mathbf{z}'} \cdot \frac{\partial \mathbf{z}'}{\partial \mathbf{x}}$$

$$= ((\hat{\mathbf{y}} - \mathbf{y}) \cdot \mathbf{W}''^\top) \odot (\mathbf{x}\mathbf{W}' + \mathbf{b}' > 0) \cdot \mathbf{W}'^\top$$

where $\odot$ represents the element-wise multiplication (Hadamard product).

When generalized for the entire dataset, the matrix ($\mathbb{R}^{N \times F}$) format becomes:

$$\nabla_{\mathbf{X}} = \frac{\partial \mathcal{L}}{\partial \mathbf{X}} = \frac{\partial \mathcal{L}}{\partial \mathbf{Z}''} \cdot \frac{\partial \mathbf{Z}''}{\partial \mathbf{Z}'} \cdot \frac{\partial \mathbf{Z}'}{\partial \mathbf{X}}$$

$$= ((\hat{\mathbf{Y}} - \mathbf{Y}) \cdot \mathbf{W}''^\top) \odot (\mathbf{X}\mathbf{W}' + \mathbf{b}' > 0) \cdot \mathbf{W}'^\top$$

$\square$

A.2 DISCUSSION ON CONVERGENCE OF COLUMN-WISE FEATURE PROPAGATION

One of the hallmark advantages of FP is its ability to guarantee convergence of feature representations for missing nodes, provided the graph is undirected and maintains strong connectivity (Berman & Plemmons, 1994). In contrast to graph domains where the initial graph structure is given without any missing elements and can thereby extract a strongly connected component, our situation, defined by initially missing features devoid of a graph structure, requires manual graph construction, such as the $k$NN graph as detailed in Equation 2. This approach does not ensure strong connectivity, making the convergence of imputed values for missing features uncertain. Nonetheless, we argue that within our context of missing features, simply increasing the number of neighbors, $k$, to achieve the convergence property might not always be advantageous.

*Claim:* Elevating $k$ to attain strong connectivity (which increases the likelihood, albeit without guarantees) and consequently secure the convergence property can sometimes be detrimental to performance. This might inadvertently introduce a primary drawback inherent to graph-based learning: over-smoothing. ⇔ *Rationale:* As the value of $k$ escalates, the adjacency matrix $\mathbf{A}^{\text{feat}}$ becomes increasingly dense. However, considering our scenario of missing features where feature representation remains incomplete, the veracity of the new connections becomes dubious. For the representation of missing nodes in the feature matrix used in Equation 2, denoted as $\mathbf{X}^{\top} \in \mathbb{R}^{F \times N}$ and represented by $\mathbf{x}_u \in \mathbb{R}^N$, a high missing rate combined with an extensive $k$ implies that the feature representation of the majority node, $\mathbf{x}_u$, will evolve via feature propagation. As the number of layers increases and $k$ approaches the total number of nodes $F$, these nodes end up with almost identical representations.

Given this perspective, we aim to avert ambiguous node connections and counteract over-smoothing, which could potentially degrade classification performance. To this end, we commit to using a relatively modest and smaller value of $k$ when crafting the graph from the feature's perspective.

**Discussion on the Convergence and Performance Gain Relationship.** To further investigate whether the convergence property contributes to performance gain, we conducted an empirical analysis to validate our claims. In Figure 7, we extended our proposed range of $k_{\text{col}}$ and $k_{\text{row}}$ values, $\{1,3,5,10\}$, up to 50, and tested the resulting graph's connectivity. We observed that when $k_{\text{col}}$ and $k_{\text{row}}$ exceed 10, the generated graph becomes strongly connected, meaning that every node is reachable from every other node. Interestingly, while strong connectivity provides convenience in choosing the number of neighbors and satisfies the necessary condition for FP to converge, it *does not necessarily translate to performance gains*. Optimal performance was, in fact, achieved within a smaller range of $k$ values, as initially proposed. Upon further investigation, we discovered that increasing $k_{\text{col}}$ leads to an *oversmoothing issue* in the resulting output, particularly in the warmed-up matrix. This effect was quantified using the MADGap metric (Chen et al., 2020a), which measures the representational difference between remote and neighboring nodes. In summary, our findings suggest that when dealing with bio-medical tabular data, where an initial graph structure is not provided and a $k$NN graph must be manually generated, selecting a large $k$ value to leverage the convergence property of FP may not be the most effective strategy in scenarios with severe missing.

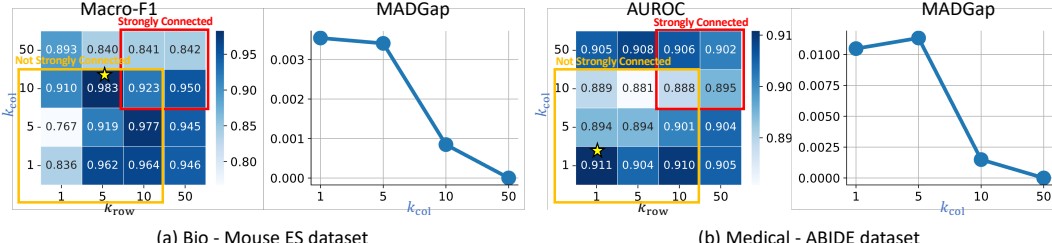

(a) Bio - Mouse ES dataset          (b) Medical - ABIDE dataset

Figure 7: Performance comparison upon increasing the values of $k_{\text{col}}$ and $k_{\text{row}}$, which are responsible for generating the column-wise and row-wise graphs, respectively. This increase ensures convergence in (a) the Mouse ES dataset and (b) a Medical dataset. For each dataset, the best-performing models, FP and NM, with GRASS initialized, are utilized to assess performance. The MADGap metric, calculated as the normalized distance in the warmed-up matrix ($\hat{\mathbf{X}}$) between remote nodes within an 8-hop distance and neighboring nodes within a 3-hop distance (as suggested in the original paper), is used to measure oversmoothing. A smaller MADGap value indicates a more severe oversmoothing.

Table 2 provides an overview of dataset statistics. In the medical domain, where features can be both numerical and categorical, we employed MinMaxScaler for numerical columns and one-hot encoding for categorical ones. For the bio domain, we employed datasets from the single-cell RNA-sequencing domain. In this domain, both false-zeros and biologically true zeros coexist (van Dijk et al., 2018; Li & Li, 2018). However, since we cannot distinguish whether a given zero is a false-zero or a true-zero, we treat this situation as a missing data scenario. Accordingly, we consider zeros as missing values, aligning with the approach taken in the recent work, scFP (Yun et al., 2023). The initial missing rate (IMR) represents the absence of data in the original table before any preprocessing. The final column of Table 2 indicates the extent of missing data even after obtaining the warmed-up feature matrix and adjacency matrix. This phenomenon is particularly evident in datasets with categorical features. Yet, the designed allowance for subsequent graph-based imputation methods has proven to complement effectively, as illustrated in Table 1. Tables 3 and 4 detail the optimal hyperparameter settings when GRASS and existing graph-based imputation models are best aligned.

Table 2: Statistics of datasets. IMR: Initially Missing Rate

| Dataset | Domain | $N$ | $F$ | Num. | Cat. | Preprocessed | $C$ | IMR | GRASS Init. |
|---|---|---|---|---|---|---|---|---|---|
| Mouse ES | Bio | 2717 | 24047 | 24047 | 0 | 2000 | 4 | 27.21% | 0.00% |
| Pancreas | Bio | 1937 | 15575 | 15575 | 0 | 2000 | 14 | 56.65% | 0.00% |
| Baron Human | Bio | 8569 | 17499 | 17499 | 0 | 2000 | 14 | 57.25% | 0.00% |
| Mouse Bladder | Bio | 2746 | 19771 | 19771 | 0 | 2000 | 16 | 69.05% | 0.14% |
| Breast Cancer | Medical | 286 | 9 | 1 | 8 | 39 | 2 | 0.35% | 0.00% |
| Hepatitis | Medical | 155 | 19 | 4 | 15 | 298 | 2 | 5.67% | 5.53% |
| Duke Breast | Medical | 907 | 93 | 34 | 59 | 3364 | 3 | 11.94% | 9.42% |
| ADNI | Medical | 2419 | 113 | 92 | 21 | 2741 | 5 | 30.02% | 4.18% |
| ABIDE | Medical | 1112 | 72 | 64 | 8 | 284 | 2 | 69.74% | 3.39% |

Table 3: Hyperparameter setting of Best Performing models.

| Dataset | Best Performing | $\theta$ | $k_{\text{col}}$ | $k_{\text{row}}$ | OG | GRASS | Improvement |
|---|---|---|---|---|---|---|---|
| Mouse ES | FP | - | 10 | 5 | 0.900 | 0.983 | 9.17% |
| Pancreas | LP | - | 3 | 3 | 0.656 | 0.799 | 21.66% |
| Baron Human | scFP | - | 1 | 10 | 0.809 | 0.853 | 5.43% |
| Mouse Bladder | PaGNN | - | 3 | 5 | 0.713 | 0.760 | 8.78% |
| Breast Cancer | GCNMF | 0.2 | 3 | 5 | 0.552 | 0.580 | 5.02% |
| Hepatitis | PaGNN | 0.6 | 5 | 1 | 0.729 | 0.742 | 1.74% |
| Duke Breast | GAIN | 0.4 | 5 | 10 | 0.699 | 0.700 | 0.09% |
| ADNI | Zero | 0.4 | 10 | 10 | 0.956 | 0.960 | 0.16% |
| ABIDE | NM | 0.2 | 1 | 3 | 0.905 | 0.919 | 1.48% |

Table 4: Hyperparameter setting of Most Improved models.

| Dataset | Most Improved | $\theta$ | $k_{\text{col}}$ | $k_{\text{row}}$ | OG | GRASS | Improvement |
|---|---|---|---|---|---|---|---|
| Mouse ES | GCNMF | - | 5 | 1 | 0.525 | 0.973 | 85.31% |
| Pancreas | GCNMF | - | 10 | 1 | 0.527 | 0.708 | 34.27% |
| Baron Human | GCNMF | - | 5 | 3 | 0.350 | 0.818 | 133.30% |
| Mouse Bladder | GCNMF | - | 1 | 3 | 0.300 | 0.702 | 133.90% |
| Breast Cancer | NM | 0.6 | 10 | 10 | 0.539 | 0.565 | 5.07% |
| Hepatitis | GAIN | 0.0 | 1 | 10 | 0.579 | 0.646 | 11.63% |
| Duke Breast | FP | 0.4 | 3 | 5 | 0.661 | 0.689 | 5.07% |
| ADNI | GCNMF | 0.0 | 3 | 10 | 0.898 | 0.945 | 5.25% |
| ABIDE | Mean | 0.2 | 5 | 10 | 0.608 | 0.906 | 49.09% |

**Preprocessing of Datasets.** In the bio datasets, we make use of cell-gene matrices to predict the relevant annotated cell types for each cell. This cell type information serves as the supervisory signal during training. For preprocessing, we typically filter out cells and genes that have not been transcribed in each row and column, respectively, and apply a log transformation to normalize the count values.

- The Mouse ES (Klein et al., 2015) dataset employs a droplet-microfluidic approach for parallel barcoding. We used concatenated data originally separated by different days post-leukemia inhibitory factor (LIF) withdrawal, treating the day of withdrawal as the annotation for the cell type.

- The Pancreas (Luecken et al., 2022) dataset, obtained via the inDrop method, captures the transcriptomes of individual pancreatic cells from four human donors and two mouse strains. It includes 14 annotated cell types.

- The Baron Human (Baron et al., 2016) dataset focuses on individual pancreatic cells from human donors, sequenced using a droplet-based method. It features 14 annotated cell types.

- The Mouse Bladder (Han et al., 2018) dataset, sourced from the Mouse Cell Atlas (MCA) project and sequenced via the Microwell-seq platform, includes cell types as defined by the original authors' annotations.

In our medical datasets, we focused on datasets that originally include missing values and feature a mix of categorical and numerical features. During preprocessing, we removed rows and columns if all features were missing in each sample or if all samples were missing in each feature, respectively. We selected the most representative feature column related to the patient's diagnosis as the class label for prediction.

- **Breast Cancer** (Asuncion & Newman, 2007): Published in the UCI repository and provided by the Oncology Institute, this dataset contains tumour-related features. We use 'recurrence', a binary attribute, as the class label.

- **Hepatitis** (Asuncion & Newman, 2007): Also published in the UCI repository, this dataset includes data on hepatitis occurrences in individuals, with attributes related to liver characteristics. The binary annotation of the patient's outcome (die or live) is used as the class label.

- **Duke Breast** (Saha et al., 2018): Made available by The Cancer Imaging Archive (TCIA), this dataset consists of medical images and non-image clinical data for tumor prediction. From the tabular data provided, we use the 'Tumor_Grade' feature, which indicates the grade of the tumor, as the class label.

- **ADNI** (Petersen et al., 2010): This collection includes various types of medical images and non-image clinical data related to Alzheimer's disease. We utilize the 'DX_bl' feature from the clinical data, indicating the patient's diagnosis, as the class label.

- **ABIDE** (Di Martino et al., 2014): Containing data on autism spectrum disorder based on brain imaging and clinical data, this dataset uses the 'DX_Group' feature from the clinical data, which represents the diagnostic group of the patient, as the class label.

**Baselines.** To tackle the challenge of generalizing graph-based imputation methods to bio-medical tabular data, we have adopted two types of baseline approaches. For graph-based imputation methods, which typically target downstream tasks like classification, we adopted widely-used methods as follows. For GNN-based methods, we utilized a 2-layer GCN as a classifier.

- LP (Zhu, 2005) is a semi-supervised algorithm that spreads known labels to similar data points in an unlabeled dataset, based on the given graph structure.

- GCNMF (Taguchi et al., 2021) is an end-to-end GNN-based model that imputes missing features by assuming a Gaussian Mixture Model aligned with GCN.

- PaGNN (Jiang & Zhang, 2020) is a GNN-based method that implements a partial message-passing scheme, propagating only observed features.

- GCN-Zero (Rossi et al., 2021) is a simple 2-layer Graph Convolution Network. We impute missing features with zeros in this model.

- GCN-NM (Rossi et al., 2021) imputes missing features by averaging the features of one-hop neighboring nodes, followed by GCN layers.
- GCN-FP (Rossi et al., 2021) propagates given features through neighbors, replacing observed ones with their original values to minimize Dirichlet energy.
- PFCI (Um et al., 2023) improves upon FP by considering propagation among feature channels with pseudo confidence, defined by the shortest path to the known feature.

Additionally, as our primary focus is on tabular data, we include common table-based imputation methods as follows.

- Mean (Little & Rubin, 2019) replaces missing values in a dataset with the mean value of the available data for the same feature.
- kNN (Troyanskaya et al., 2001) imputes missing data by finding the $k$ nearest neighbors based on cosine similarity and then averaging their features.
- GAIN (Yoon et al., 2018) uses a generative adversarial network to impute missing values, where one network generates candidates and another evaluates them.
- MIWAE (Mattei & Frellsen, 2019) employs a type of autoencoder for multiple imputations, capturing the data's underlying distribution to provide multiple plausible values for missing data.
- GRAPE (You et al., 2020) adopts a bipartite graph framework, viewing observations and features as two node types, and imputes missing values through edge-level prediction.
- IGRM (Zhong et al., 2023) enhances the bipartite graph framework by introducing the concept of a friend network, which denotes relationships between samples.

## A.4   PSEUDOCODE OF GRASS

Algorithm 1 presents the pseudocode for our proposed algorithm, GRASS. By training the MLP, we derive the feature gradient, which is utilized to generate a column-wise graph (see line 1). We then execute Column-wise Feature Propagation (line 3) and clamp the categorical columns (line 4). Consequently, we produce the warmed-up feature matrix and the adjacency matrix, which will seamlessly align with existing graph-based imputation methods in lines 5 and 6, respectively. It's worth noting that the feature gradient mentioned in line 13 can be readily obtained using AutoGrad in PyTorch (Paszke et al., 2017). More precisely, we provide a PyTorch-style pseudocode in Listing 1, detailing the function for obtaining the feature gradient. In training the 2-layer MLP, as shown in Line 21, we activate the 'requires_grad' attribute by setting it to True. This enables AutoGrad in PyTorch to automatically calculate the feature gradient following backpropagation, a value that is then accessible in Line 28. It is crucial to note that there is no update to the original feature matrix; it remains static, with only the classifier's weights being updated. This process dynamically alters the value of the feature gradient through these modified weights, as demonstrated in Proposition 1. Additionally, as indicated in Line 37, we save the feature gradient only when there is an improvement in validation performance, which is an efficient approach to memory usage. After training the MLP, which typically involves early stopping, we compute the average of the gradients to obtain the final feature gradient.

---

**Algorithm 1** Pseudocode of the proposed algorithm

---

  **Input:** Initially missing feature matrix $\mathbf{X}$, train label matrix $\mathbf{Y}$
  **Output:** Warmed-up feature matrix $\hat{\mathbf{X}}$, adjacency matrix $\hat{\mathbf{A}}$
1: $\overline{\boldsymbol{\nabla}}_{\mathbf{X}} \leftarrow \text{TRAINMLP}(\mathbf{X}, ValidationSet)$        ▷ Train MLP & Obtain Feature Gradient
2: $\mathbf{A}_{feat} \leftarrow k_{\text{col}}\text{-nearest-neighbor}(\overline{\boldsymbol{\nabla}}_{\mathbf{X}}^{\top} \| \mathbf{X}^{\top})$        ▷ Column-wise Graph Generation
3: $\mathbf{X}^{(K)\top} \leftarrow \text{PROPAGATION}(\mathbf{A}_{feat}, \mathbf{X}^{(0)\top}, \mathcal{V}_{known}, K)$        ▷ Column-wise FP
4: $\hat{\mathbf{X}} \leftarrow \text{CLAMPER}(\mathbf{X}^{(K)\top})$        ▷ Clamping categorical columns
5: $\hat{\mathbf{A}} \leftarrow k_{\text{row}}\text{-nearest-neighbor}(\hat{\mathbf{X}})$
6: **function** TRAINMLP($\mathbf{X}, ValidationSet$)
7:      Initialize highest validation performance as $V_{\text{highest}} = 0$
8:      Initialize empty list $G = []$
9:      **while** not converged **do**
10:          Train MLP for one epoch using training data
11:          Compute validation performance $V_{\text{current}}$
12:          **if** $V_{\text{current}} > V_{\text{highest}}$ **then**
13:              $\boldsymbol{\nabla}_{\mathbf{X}} \leftarrow ((\hat{\mathbf{Y}} - \mathbf{Y}) \cdot \mathbf{W}''^{\top}) \odot (\mathbf{X}\mathbf{W}' + \mathbf{b}' > 0) \cdot \mathbf{W}'^{\top}$
14:              Append the $\boldsymbol{\nabla}_{\mathbf{X}}$ to list $G$
15:              Update $V_{\text{highest}} \leftarrow V_{\text{current}}$
16:          **end if**
17:      **end while**
18:      $\overline{\boldsymbol{\nabla}}_{\mathbf{X}} \leftarrow \frac{1}{\text{length}(G)} \sum_{g \in G} g$
19:      **return** $\overline{\boldsymbol{\nabla}}_{\mathbf{X}}$
20: **end function**
21: **function** PROPAGATION($\mathbf{A}, \mathbf{W}, Known, K$)
22:      $\mathbf{M} \leftarrow \mathbf{W}$
23:      **for** $k \leftarrow 1$ to $K$ **do**
24:          $\mathbf{W} \leftarrow \mathbf{A}\mathbf{W}$
25:          $\mathbf{W}_{Known} \leftarrow \mathbf{M}_{Known}$
26:      **end for**
27:      **return** $\mathbf{W}$
28: **end function**
29: **function** CLAMPER($\hat{\mathbf{X}}$)
30:      **for** $i \leftarrow 1$ to $N$ **do**
31:          **for** $j \leftarrow c$ to length($CategoricalColumns$) **do**
32:              $\tilde{\mathbf{x}}_c \leftarrow \text{softmax}(\hat{\mathbf{X}}_{j,c:c+c_b})$
33:              $\hat{\mathbf{X}}_{j,c:c+c_b} = \begin{cases} \text{OneHot}(\text{argmax}(\tilde{\mathbf{x}}_c)), & \text{if } \max(\tilde{\mathbf{x}}_c) \geq \theta \\ \underbrace{[?, \ldots, ?]}_{c_b \text{ times}}, & \text{otherwise} \end{cases}$
34:          **end for**
35:      **end for**
36:      **return** $\hat{\mathbf{X}}$
37: **end function**

---

```python
def obtain_feature_gradient(
            x, # initially missing feature matrix
            classifier, # 2-layer MLP
            labels, # supervisions
            train_mask, # indices for training set
            val_mask, # indices for validation set
            epochs # training epochs
            )

    # Initialize missing features as zeros
    x = torch.nan_to_num(x, 0)

    optimizer = optim.Adam(classifier.parameters())
    best_val_performance = 0
    grads = []

    for epoch in range(0, epochs):
        classifier.train()
        optimizer.zero_grad()

        x.requires_grad=True # allow tracking gradients for x
        out = classifier(x)
        loss = F.CrossEntropy(out[train_mask], labels[train_mask])

        loss.backward()
        optimizer.step()

        grad = x.grad # automatically calculates Proposition 1.
        x.requires_grad=False

        classifier.eval()
        out = classifier(x)

        val_performance = roc_auc_score(out[val_mask], labels[val_mask])

        # Save gradient if validation performance improves
        if best_val_performance <= val_performance:
            best_val_performance = val_performance
            grads.append(F.normalize(grad, dim=0, p=2).cpu())

    # Average gradients
    feature_gradient = torch.mean(torch.stack(grads), dim=0)

    return feature_gradient
```

Listing 1: PyTorch-style pseudocode for obtaining feature gradient via training 2-layer MLP.

## A.5 Classification Performance on Table Format

Figure 3 illustrates the performance of various baselines across four biological and five medical datasets, highlighting the enhancement achieved when utilizing GRASS for initialization. In this context, we provide a detailed numerical analysis to quantify the performance gains and their relative improvements. For the biological datasets (Mouse ES, Pancreas, Baron Human, Mouse Bladder), Macro-F1 scores are employed as the performance metric, whereas AUROC scores are used for the medical datasets (Breast Cancer, Hepatitis, Duke Breast, ADNI, ABIDE). In each table, the best-performing model's performance is bolded while the most improved model's performance is underlined.

Table 5: Bio-Mouse ES.

| | Mouse ES (IMR: 27.21%) | | |
| | OG | + GRASS init. | Impr. (%) |
|---|---|---|---|
| LP | $0.878_{\pm 0.005}$ | $0.979_{\pm 0.003}$ | 11.43 |
| GCNMF | $0.525_{\pm 0.238}$ | $0.972_{\pm 0.008}$ | 85.31 |
| PaGNN | $0.899_{\pm 0.072}$ | $0.980_{\pm 0.002}$ | 9.03 |
| GCN-zero | $0.960_{\pm 0.005}$ | $0.982_{\pm 0.004}$ | 2.30 |
| GCN-nm | $0.885_{\pm 0.098}$ | $0.982_{\pm 0.004}$ | 10.99 |
| GCN-FP | $0.900_{\pm 0.100}$ | $\mathbf{0.982}_{\pm 0.003}$ | 9.17 |
| PCFI | $0.949_{\pm 0.004}$ | $0.955_{\pm 0.006}$ | 0.57 |
| Mean | $0.979_{\pm 0.006}$ | $0.979_{\pm 0.004}$ | 0.08 |
| kNN | $0.969_{\pm 0.011}$ | $0.977_{\pm 0.005}$ | 0.83 |
| GAIN | $0.978_{\pm 0.011}$ | $0.982_{\pm 0.007}$ | 0.39 |
| MIWAE | OOM | - | - |
| GRAPE | OOM | - | - |
| IGRM | OOM | - | - |
| scFP | $0.952_{\pm 0.004}$ | $0.976_{\pm 0.003}$ | 2.52 |

Table 6: Bio-Pancreas.

| | Pancreas (IMR: 56.65%) | | |
| | OG | + GRASS init. | Impr. (%) |
|---|---|---|---|
| LP | $0.656_{\pm 0.039}$ | $\mathbf{0.798}_{\pm 0.068}$ | 21.66 |
| GCNMF | $0.527_{\pm 0.210}$ | $0.708_{\pm 0.087}$ | 34.27 |
| PaGNN | $0.701_{\pm 0.044}$ | $0.768_{\pm 0.040}$ | 9.58 |
| GCN-zero | $0.687_{\pm 0.066}$ | $0.783_{\pm 0.062}$ | 14.02 |
| GCN-nm | $0.679_{\pm 0.047}$ | $0.788_{\pm 0.068}$ | 16.09 |
| GCN-FP | $0.716_{\pm 0.046}$ | $0.788_{\pm 0.068}$ | 10.08 |
| PCFI | $0.673_{\pm 0.055}$ | $0.686_{\pm 0.040}$ | 1.95 |
| Mean | $0.616_{\pm 0.044}$ | $0.619_{\pm 0.032}$ | 0.44 |
| kNN | $0.652_{\pm 0.047}$ | $0.706_{\pm 0.048}$ | 8.34 |
| GAIN | $0.638_{\pm 0.075}$ | $0.738_{\pm 0.024}$ | 15.66 |
| MIWAE | OOM | - | - |
| GRAPE | OOM | - | - |
| IGRM | OOM | - | - |
| scFP | $0.743_{\pm 0.044}$ | $0.788_{\pm 0.085}$ | 6.05 |

Table 7: Bio-Baron Human.

| | Baron Human (IMR: 57.25%) | | |
| | OG | + GRASS init. | Impr. (%) |
|---|---|---|---|
| LP | $0.736_{\pm 0.022}$ | $0.828_{\pm 0.055}$ | 12.46 |
| GCNMF | $0.350_{\pm 0.130}$ | $0.817_{\pm 0.066}$ | 133.30 |
| PaGNN | $0.777_{\pm 0.043}$ | $0.820_{\pm 0.057}$ | 5.53 |
| GCN-zero | $0.812_{\pm 0.030}$ | $0.842_{\pm 0.049}$ | 3.71 |
| GCN-nm | $0.758_{\pm 0.045}$ | $0.801_{\pm 0.084}$ | 5.71 |
| GCN-FP | $0.789_{\pm 0.039}$ | $0.802_{\pm 0.084}$ | 1.61 |
| PCFI | $0.769_{\pm 0.036}$ | $0.792_{\pm 0.038}$ | 2.96 |
| Mean | $0.672_{\pm 0.010}$ | $0.694_{\pm 0.023}$ | 3.41 |
| kNN | $0.746_{\pm 0.048}$ | $0.760_{\pm 0.053}$ | 1.82 |
| GAIN | $0.728_{\pm 0.041}$ | $0.745_{\pm 0.033}$ | 2.39 |
| MIWAE | OOM | - | - |
| GRAPE | OOM | - | - |
| IGRM | OOM | - | - |
| scFP | $0.809_{\pm 0.067}$ | $\mathbf{0.853}_{\pm 0.0.031}$ | 5.43 |

Table 8: Bio-Mouse Bladder.

| | Mouse Bladder (IMR: 69.05%) | | |
| | OG | + GRASS init. | Impr. (%) |
|---|---|---|---|
| LP | $0.556_{\pm 0.030}$ | $0.643_{\pm 0.053}$ | 15.57 |
| GCNMF | $0.300_{\pm 0.182}$ | $0.701_{\pm 0.042}$ | 133.90 |
| PaGNN | $0.713_{\pm 0.056}$ | $\mathbf{0.775}_{\pm 0.028}$ | 8.78 |
| GCN-zero | $0.712_{\pm 0.015}$ | $0.768_{\pm 0.031}$ | 7.83 |
| GCN-nm | $0.721_{\pm 0.050}$ | $0.775_{\pm 0.030}$ | 7.38 |
| GCN-FP | $0.686_{\pm 0.048}$ | $0.772_{\pm 0.036}$ | 12.48 |
| PCFI | $0.710_{\pm 0.046}$ | $0.727_{\pm 0.028}$ | 2.41 |
| Mean | $0.555_{\pm 0.074}$ | $0.569_{\pm 0.062}$ | 2.39 |
| kNN | $0.587_{\pm 0.038}$ | $0.674_{\pm 0.059}$ | 14.82 |
| GAIN | $0.585_{\pm 0.030}$ | $0.649_{\pm 0.038}$ | 10.84 |
| MIWAE | OOM | - | - |
| GRAPE | OOM | - | - |
| IGRM | OOM | - | - |
| scFP | $0.653_{\pm 0.024}$ | $0.759_{\pm 0.022}$ | 16.23 |

Table 9: Medical-Breast Cancer.

| | Breast Cancer (IMR: 0.35%) | | |
| | OG | + GRASS init. | Impr. (%) |
|---|---|---|---|
| LP | $0.561_{\pm 0.038}$ | $0.562_{\pm 0.041}$ | 0.14 |
| GCNMF | $0.551_{\pm 0.033}$ | $\mathbf{0.579}_{\pm 0.049}$ | 5.02 |
| PaGNN | $0.540_{\pm 0.037}$ | $0.562_{\pm 0.032}$ | 3.98 |
| GCN-zero | $0.542_{\pm 0.048}$ | $0.557_{\pm 0.039}$ | 2.71 |
| GCN-nm | $0.538_{\pm 0.049}$ | $0.566_{\pm 0.052}$ | 5.07 |
| GCN-FP | $0.543_{\pm 0.047}$ | $0.565_{\pm 0.052}$ | 4.05 |
| PCFI | $0.545_{\pm 0.039}$ | $0.547_{\pm 0.040}$ | 0.44 |
| Mean | $0.562_{\pm 0.045}$ | $0.562_{\pm 0.045}$ | 0.00 |
| kNN | $0.552_{\pm 0.041}$ | $0.556_{\pm 0.041}$ | 0.67 |
| GAIN | $0.566_{\pm 0.044}$ | $0.567_{\pm 0.043}$ | 0.21 |
| MIWAE | $0.558_{\pm 0.033}$ | $0.563_{\pm 0.035}$ | 0.93 |
| GRAPE | $0.572_{\pm 0.029}$ | $0.573_{\pm 0.017}$ | 0.26 |
| IGRM | $0.548_{\pm 0.039}$ | $0.552_{\pm 0.037}$ | 0.66 |
| scFP | $0.554_{\pm 0.047}$ | $0.563_{\pm 0.055}$ | 1.62 |

Table 10: Medical-Hepatitis.

| | Hepatitis (IMR: 5.67%) | | |
| | OG | + GRASS init. | Impr. (%) |
|---|---|---|---|
| LP | $0.573_{\pm 0.078}$ | $0.608_{\pm 0.053}$ | 6.06 |
| GCNMF | $0.685_{\pm 0.097}$ | $0.707_{\pm 0.088}$ | 3.22 |
| PaGNN | $0.729_{\pm 0.074}$ | $\mathbf{0.741}_{\pm 0.058}$ | 1.74 |
| GCN-zero | $0.713_{\pm 0.090}$ | $0.714_{\pm 0.088}$ | 0.14 |
| GCN-nm | $0.702_{\pm 0.071}$ | $0.702_{\pm 0.071}$ | 0.00 |
| GCN-FP | $0.705_{\pm 0.045}$ | $0.707_{\pm 0.092}$ | 0.20 |
| PCFI | $0.728_{\pm 0.108}$ | $0.728_{\pm 0.108}$ | 0.00 |
| Mean | $0.691_{\pm 0.072}$ | $0.711_{\pm 0.081}$ | 2.86 |
| kNN | $0.612_{\pm 0.097}$ | $0.626_{\pm 0.105}$ | 2.15 |
| GAIN | $0.578_{\pm 0.093}$ | $0.646_{\pm 0.080}$ | 11.63 |
| MIWAE | $0.573_{\pm 0.080}$ | $0.608_{\pm 0.077}$ | 6.25 |
| GRAPE | $0.701_{\pm 0.033}$ | $0.706_{\pm 0.032}$ | 0.63 |
| IGRM | $0.668_{\pm 0.087}$ | $0.703_{\pm 0.109}$ | 5.26 |
| scFP | $0.691_{\pm 0.077}$ | $0.691_{\pm 0.077}$ | 0.00 |

Table 11: Medical-Duke Breast.

| | Duke Breast (IMR: 11.94%) | | |
| | OG | + GRASS init. | Impr. (%) |
|---|---|---|---|
| LP | $0.672_{\pm 0.021}$ | $0.678_{\pm 0.026}$ | 0.98 |
| GCNMF | $0.664_{\pm 0.035}$ | $0.688_{\pm 0.032}$ | 3.61 |
| PaGNN | $0.685_{\pm 0.033}$ | $0.690_{\pm 0.029}$ | 0.69 |
| GCN-zero | $0.673_{\pm 0.022}$ | $0.694_{\pm 0.021}$ | 3.13 |
| GCN-nm | $0.678_{\pm 0.033}$ | $0.691_{\pm 0.025}$ | 1.96 |
| GCN-FP | $0.661_{\pm 0.031}$ | $0.688_{\pm 0.028}$ | 4.21 |
| PCFI | $0.693_{\pm 0.029}$ | $0.696_{\pm 0.030}$ | 0.40 |
| Mean | $0.687_{\pm 0.018}$ | $0.687_{\pm 0.019}$ | 0.04 |
| kNN | $0.692_{\pm 0.026}$ | $0.697_{\pm 0.014}$ | 0.74 |
| GAIN | $0.699_{\pm 0.018}$ | $\mathbf{0.699}_{\pm 0.017}$ | 0.09 |
| MIWAE | $0.692_{\pm 0.013}$ | $0.693_{\pm 0.012}$ | 0.13 |
| GRAPE | OOM | - | - |
| IGRM | OOM | - | - |
| scFP | $0.678_{\pm 0.031}$ | $0.690_{\pm 0.030}$ | 1.76 |

Table 12: Medical-ADNI.

| | ADNI (IMR: 30.02%) | | |
| | OG | + GRASS init. | Impr. (%) |
|---|---|---|---|
| LP | $0.928_{\pm 0.005}$ | $0.943_{\pm 0.005}$ | 1.56 |
| GCNMF | $0.897_{\pm 0.045}$ | $0.944_{\pm 0.004}$ | 5.25 |
| PaGNN | $0.953_{\pm 0.003}$ | $0.955_{\pm 0.003}$ | 0.27 |
| GCN-zero | $0.956_{\pm 0.003}$ | $\mathbf{0.957}_{\pm 0.003}$ | 0.17 |
| GCN-nm | $0.955_{\pm 0.003}$ | $0.956_{\pm 0.003}$ | 0.19 |
| GCN-FP | $0.955_{\pm 0.003}$ | $0.957_{\pm 0.003}$ | 0.18 |
| PCFI | $0.951_{\pm 0.004}$ | $0.955_{\pm 0.003}$ | 0.46 |
| Mean | $0.939_{\pm 0.002}$ | $0.943_{\pm 0.003}$ | 0.46 |
| kNN | $0.943_{\pm 0.003}$ | $0.943_{\pm 0.004}$ | 0.01 |
| GAIN | $0.937_{\pm 0.003}$ | $0.944_{\pm 0.003}$ | 0.67 |
| MIWAE | OOM | - | - |
| GRAPE | OOM | - | - |
| IGRM | OOM | - | - |
| scFP | $0.953_{\pm 0.003}$ | $0.954_{\pm 0.002}$ | 0.10 |

Table 13: Medical-ABIDE.

| | ABIDE (IMR: 69.74%) | | |
| | OG | + GRASS init. | Impr. (%) |
|---|---|---|---|
| LP | $0.894_{\pm 0.009}$ | $0.895_{\pm 0.011}$ | 0.13 |
| GCNMF | $0.819_{\pm 0.042}$ | $0.913_{\pm 0.010}$ | 11.49 |
| PaGNN | $0.907_{\pm 0.009}$ | $0.914_{\pm 0.008}$ | 0.82 |
| GCN-zero | $0.902_{\pm 0.008}$ | $0.915_{\pm 0.008}$ | 1.38 |
| GCN-nm | $0.905_{\pm 0.011}$ | $\mathbf{0.918}_{\pm 0.007}$ | 1.48 |
| GCN-FP | $0.908_{\pm 0.014}$ | $0.915_{\pm 0.005}$ | 0.86 |
| PCFI | $0.915_{\pm 0.008}$ | $0.917_{\pm 0.010}$ | 0.26 |
| Mean | $0.607_{\pm 0.027}$ | $0.905_{\pm 0.007}$ | 49.09 |
| kNN | $0.896_{\pm 0.009}$ | $0.907_{\pm 0.010}$ | 1.16 |
| GAIN | $0.793_{\pm 0.010}$ | $0.910_{\pm 0.009}$ | 14.70 |
| MIWAE | $0.623_{\pm 0.015}$ | $0.898_{\pm 0.008}$ | 44.10 |
| GRAPE | $0.889_{\pm 0.010}$ | $0.906_{\pm 0.006}$ | 1.90 |
| IGRM | $0.747_{\pm 0.019}$ | $0.908_{\pm 0.004}$ | 21.54 |
| scFP | $0.894_{\pm 0.010}$ | $0.903_{\pm 0.007}$ | 1.00 |

### A.6 Discussion Comparing scFP with GRASS

As GRASS integrates FP with the aim of enhancing generalizability in the bio-medical domain, it is necessary to compare it with the recently proposed single-cell Feature Propagation (scFP), which also adopts FP, specifically targeting the single-cell RNA-seq domain.

- **(1) Target Domain**: While scFP focuses on the single-cell RNA-seq (scRNA-seq) domain, particularly from a biological perspective, GRASS adopts a more general approach for the broader 'bio-medical' domain, as indicated in the paper's title. This distinction is crucial as scRNA-seq datasets typically comprise *numerical features* where each element represents the count of a gene's RNA transcript sequenced by the sequencing machine. In contrast, medical datasets often include *both numerical and categorical features*, such as patient information. This versatility underscores the broader applicability of GRASS, capable of handling both numerical and categorical features, the latter through the clamping technique as discussed in Section 3.3. Therefore, we argue that the target domain of scFP, primarily focused on numerical matrix imputation in scRNA-seq, differs from that of GRASS, which extends to handling categorical data often encountered in patient data.

- **(2) Target Task and Imputation Methodology**: Unlike scFP, which is *unsupervised* with its primary goal being effective imputation in sparse and noisy cell-gene count matrices, this work concentrates on *supervised* tasks, specifically on downstream applications like *classification*. Notably, the objective of imputation is often to enhance performance in relevant downstream tasks (Rossi et al., 2021; van Dijk et al., 2018; Wang et al., 2021). In this context, while the unsupervised approach of scFP can align with supervised tasks through probing (i.e., attaching a classifier), it is important to note that since its imputation occurs prior to probing, scFP cannot incorporate any downstream task-related knowledge during the imputation process, potentially leading to shortcomings in classification tasks. Conversely, as GRASS is directly designed with downstream tasks in mind, it incorporates knowledge pertinent to these tasks during imputation. This is achieved by utilizing the *feature gradient*, which is obtained during training 2-layer MLP. This fundamental difference in the target task (classification vs. imputation) and the imputation process (incorporating relevant downstream knowledge or not) distinctly sets the two methodologies apart.

- **(3) Usage of FP**: Although both scFP and GRASS employ FP, their applications of this process differ significantly. Specifically, scFP utilizes FP from a row-wise perspective, i.e., focusing on cell-cell relationships while assuming gene-gene relationship independence. Although beneficial for smoothing similar and relevant samples, this approach does not capture interactions between columns (features), which are pivotal in the bio-medical domain. For instance, in scRNA-seq, gene-gene relationships, such as co-expression networks, play a critical role in identifying key regulatory genes or pathways, offering insights into underlying biological or disease mechanisms (Cochain et al., 2018; Chowdhury et al., 2019; Galfre et al., 2021). Acknowledging this, GRASS initially employs column-wise FP to capture potential feature interactions, e.g., gene-gene relationships. It's also noteworthy that GRASS incorporates not only the feature matrix but also the feature gradient relevant to downstream tasks when generating the column-wise $k$NN graph. Consequently, before initiating row-wise (sample-wise) smoothing in the relevant GNN model, GRASS is able to consider feature relationships that scFP does not capture. This distinction is illustrated in Figure 8 and significantly differentiates the two methodologies.

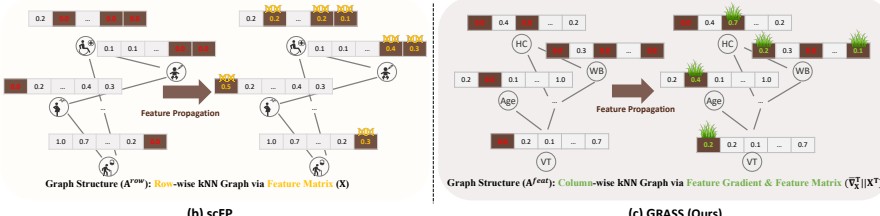

Figure 8: Comparison of scFP and GRASS. (a) scFP constructs a Row-wise $k$NN graph using only the input feature matrix ($\mathbf{X}$). In contrast, (b) GRASS constructs a Column-wise $k$NN graph incorporating both the input feature matrix ($\mathbf{X}$) and the supplementary feature gradient ($\overline{\nabla}_{\mathbf{X}}$).

## A.7 DISCUSSION ON THE COMPUTATIONAL DEMAND OF GRASS

As GRASS serves as a preprocessing step that can be integrated with existing baselines to enhance their performance, it is crucial to consider its computational demand alongside its performance benefits in two perspectives: **Memory** and **Time**.

- **Memory cost**: From a memory perspective, the primary resource utilized by GRASS is the *feature gradient* ($\overline{\nabla}_{\mathbf{X}} \in \mathbb{R}^{N \times F}$), which plays a supplemental role in constructing a column-wise graph. This feature gradient shares the same shape as the original feature matrix, with dimensions corresponding to the total number of nodes ($N$) and features ($F$). However, it is important to note that in the context of graph-based imputation models, which inherently employ a row-wise (sample-wise) adjacency matrix ($\mathbf{A} \in \mathbb{R}^{N \times N}$), the complexity associated with the adjacency matrix often surpasses that of the feature matrix, i.e., $\mathcal{O}(NF) + \mathcal{O}(N^2) = \mathcal{O}(N^2)$. This is particularly true in the bio-medical domain where datasets are typically tabular and the number of samples significantly exceeds the number of features ($N >> F$). Therefore, the additional memory requirement for storing the feature gradient is not prohibitively large. Furthermore, the complexity of the generated column-wise graph ($\mathbf{A}^{feat} \in \mathbb{R}^{F \times F}$) is also lower compared to the row-wise adjacency matrix, allowing GRASS to align with existing graph-based models without incurring excessive memory costs. Once the warmed-up matrix ($\hat{\mathbf{X}}$) and adjacency matrix ($\hat{\mathbf{A}}$) are computed, the memory allocated for the feature gradient and column-wise graph can be released, leaving only the cost of training the original baseline model for the downstream task.

- **Time cost**: From a time complexity perspective, the process is almost identical to training a conventional 2-layer MLP, which is efficient for tabular data and involves training two weight matrices: one that transforms the raw feature space to a hidden space, and another that maps the hidden space to the output space for final predictions. Despite the apparent complexity of calculating the feature gradient as outlined in Proposition 1, the actual computation, as demonstrated in Listing 1, is straightforward in terms of implementation. By enabling the 'requires_grad' switch, the gradient information is automatically saved, making the time complexity for computing the feature gradient equivalent to training a 2-layer MLP. Additionally, the column-wise Feature Propagation can be efficiently executed via sparse multiplication of the adjacency matrix and the feature matrix, as detailed in (Rossi et al., 2021). Thus, the overall time required to obtain the warmed-up matrix and adjacency matrix is not substantial.

## A.8 FURTHER EXTENSION AND GENERALIZABILITY OF GRASS

To explore the scalability of GRASS to larger datasets, we conducted evaluations using the single-cell RNA-seq Macosko dataset, which comprises 44,808 cells, 22,452 genes, and 14 distinct cell types, with an initial missing rate of 81.41%. Among these genes, we preprocessed 2,000 highly variable genes, a common technique in scRNA-seq (Yun et al., 2023). We noted that GRASS integrates smoothly with existing methods, except in cases where initial baselines, such as GCNMF and GRAPE, encounter Out-Of-Memory (OOM) issues due to the weights associated with the Gaussian Mixture Model and the construction of a heterogeneous node-feature graph, respectively. In Table 14, it is observed that graph-based methods can enhance their performance when combined with GRASS. In large graphs, since the feature dimension typically does not surpass the number of samples (which is usually the case), GRASS aligns well with current graph-based imputation methods.

Additionally, while GRASS is primarily designed for the bio-medical domain, we also assessed its applicability to other domains. For this purpose, we utilized the Wine dataset (Asuncion & Newman, 2007), which consists of 178 samples with 14 numerical features and 3 classes. As the Wine dataset initially lacks missing values, we introduced a 30% uniform missing scenario by manually dropping features. Table 15 demonstrates that using GRASS as an initializer, enabling existing models to start with a warmed-up feature matrix and adjacency matrix, effectively benefits other domains as well. This highlights the potential of GRASS for broader generalizability beyond the bio-medical sphere.

Table 14: Scalability-Macosko dataset.

| | Macosko (IMR: 81.41%) | | |
| --- | --- | --- | --- |
| | OG | + GRASS init. | Impr. (%) |
| LP | $0.853_{\pm0.025}$ | $\underline{0.870}_{\pm0.025}$ | 7.19 |
| GCNMF | OOM | - | - |
| PaGNN | $0.938_{\pm0.008}$ | $0.939_{\pm0.001}$ | 0.17 |
| GCN-zero | $0.920_{\pm0.031}$ | $0.929_{\pm0.006}$ | 0.92 |
| GCN-nm | $0.923_{\pm0.017}$ | $0.930_{\pm0.071}$ | 0.71 |
| GCN-FP | $0.937_{\pm0.006}$ | $0.941_{\pm0.045}$ | 0.43 |
| PCFI | $0.932_{\pm0.017}$ | $0.939_{\pm0.005}$ | 0.75 |
| Mean | $0.819_{\pm0.042}$ | $0.835_{\pm0.048}$ | 1.87 |
| kNN | $0.904_{\pm0.021}$ | $0.910_{\pm0.012}$ | 0.62 |
| GAIN | $0.891_{\pm0.039}$ | $0.898_{\pm0.012}$ | 0.86 |
| MIWAE | OOM | - | - |
| GRAPE | OOM | - | - |
| IGRM | OOM | - | - |
| scFP | $0.934_{\pm0.011}$ | $\mathbf{0.941}_{\pm0.020}$ | 0.72 |

Table 15: Generalizability-Wine dataset.

| | Wine (IMR: 0.00%) | | |
| --- | --- | --- | --- |
| | OG | + GRASS init. | Impr. (%) |
| LP | $0.647_{\pm0.017}$ | $0.647_{\pm0.017}$ | 0.00 |
| GCNMF | $0.656_{\pm0.027}$ | $0.657_{\pm0.023}$ | 0.11 |
| PaGNN | $0.650_{\pm0.030}$ | $0.661_{\pm0.023}$ | 1.65 |
| GCN-zero | $0.637_{\pm0.043}$ | $0.648_{\pm0.033}$ | 1.68 |
| GCN-nm | $0.629_{\pm0.034}$ | $0.660_{\pm0.030}$ | 4.93 |
| GCN-FP | $0.642_{\pm0.032}$ | $0.647_{\pm0.042}$ | 0.79 |
| PCFI | $0.650_{\pm0.042}$ | $\mathbf{0.670}_{\pm0.031}$ | 3.06 |
| Mean | $0.585_{\pm0.028}$ | $0.600_{\pm0.021}$ | 2.46 |
| kNN | $0.629_{\pm0.012}$ | $0.640_{\pm0.012}$ | 1.75 |
| GAIN | $0.618_{\pm0.012}$ | $0.640_{\pm0.018}$ | 3.61 |
| MIWAE | $0.514_{\pm0.027}$ | $\underline{0.591}_{\pm0.024}$ | 14.93 |
| GRAPE | $0.567_{\pm0.064}$ | $0.587_{\pm0.045}$ | 3.52 |
| IGRM | $0.573_{\pm0.022}$ | $0.579_{\pm0.044}$ | 0.96 |
| scFP | $0.620_{\pm0.022}$ | $0.620_{\pm0.026}$ | 0.10 |

## A.9 FLEXIBILITY IN SETTING HYPERPARAMETERS IN GRASS

Considering the diversity and complexity inherent in machine learning, optimal hyperparameter selection can significantly enhance model training for downstream tasks. Specifically, crucial hyperparameters like dimension size and dropout rate play a vital role in determining a model's stability and effectiveness. In this context, we conducted a sensitivity analysis on hyperparameters to ascertain if flexibility in their selection yields benefits in subsequent tasks. We explored dimension sizes of 16, 64, and 256, alongside dropout rates of 0.0, 0.25, and 0.5. Our findings, as detailed in Table 16, reveal that a hidden dimension size of 256 with a dropout rate of 0.25 is more advantageous than the conventional setting of a 64-dimensional hidden layer with a 0.5 dropout rate. Furthermore, in Table 17 and Table 18, case of an imputation model equipped solely with a logistic classifier (i.e., a 1-layer MLP) and thus invariant to hyperparameter flexibility, we found performance gains when using our proposed method as an initializer. This improvement is attributed to the trainable parameters in a 2-layer MLP and the incorporation of feature gradient. In summary, while setting constant hyperparameters ensures reproducibility and consistency across different datasets in the biomedical domain, allowing flexibility in these parameters can potentially enhance overall performance.

Table 16: Pancreas-GCNMF.

| Pancreas (IMR: 56.65%) | | |
|---|---|---|
| GCNMF | + GRASS init. | Impr. (%) |
| D: 16 / do: 0.0   $0.173_{\pm0.129}$ | $0.144_{\pm0.100}$ | -16.76 |
| D: 16 / do: 0.25   $0.058_{\pm0.007}$ | $0.329_{\pm0.252}$ | 467.24 |
| D: 16 / do: 0.5   $0.098_{\pm0.054}$ | $0.340_{\pm0.296}$ | 246.94 |
| D: 64 / do: 0.0   $0.671_{\pm0.056}$ | $0.680_{\pm0.067}$ | 1.34 |
| D: 64 / do: 0.25   $0.360_{\pm0.247}$ | $0.662_{\pm0.048}$ | 83.89 |
| D: 64 / do: 0.5   $0.443_{\pm0.026}$ | $0.677_{\pm0.065}$ | 52.82 |
| D: 256 / do: 0.0   $0.603_{\pm0.008}$ | $0.671_{\pm0.053}$ | 11.28 |
| D: 256 / do: 0.25   $0.611_{\pm0.009}$ | $\mathbf{0.692}_{\pm0.049}$ | 13.26 |
| D: 256 / do: 0.5   $0.617_{\pm0.023}$ | $0.667_{\pm0.051}$ | 8.10 |

Table 17: Pancreas-Mean.

| Pancreas (IMR: 56.65%) | | |
|---|---|---|
| Mean | + GRASS init. | Impr. (%) |
| D: 16 / do: 0.0   $0.564_{\pm0.004}$ | $0.627_{\pm0.044}$ | 11.17 |
| D: 16 / do: 0.25   $0.564_{\pm0.004}$ | $0.636_{\pm0.048}$ | 12.77 |
| D: 16 / do: 0.5   $0.564_{\pm0.004}$ | $0.613_{\pm0.026}$ | 8.69 |
| D: 64 / do: 0.0   $0.564_{\pm0.004}$ | $0.617_{\pm0.014}$ | 9.40 |
| D: 64 / do: 0.25   $0.564_{\pm0.004}$ | $0.617_{\pm0.014}$ | 9.40 |
| D: 64 / do: 0.5   $0.564_{\pm0.004}$ | $0.624_{\pm0.039}$ | 10.64 |
| D: 256 / do: 0.0   $0.564_{\pm0.004}$ | $0.639_{\pm0.056}$ | 13.30 |
| D: 256 / do: 0.25   $0.564_{\pm0.004}$ | $0.638_{\pm0.052}$ | 13.12 |
| D: 256 / do: 0.5   $0.564_{\pm0.004}$ | $\underline{0.640}_{\pm0.064}$ | 13.48 |

Table 18: Pancreas-scFP.

| Pancreas (IMR: 56.65%) | | |
|---|---|---|
| scFP | + GRASS init. | Impr. (%) |
| D: 16 / do: 0.0   $0.748_{\pm0.055}$ | $0.764_{\pm0.056}$ | 2.14 |
| D: 16 / do: 0.25   $0.748_{\pm0.055}$ | $0.764_{\pm0.059}$ | 2.14 |
| D: 16 / do: 0.5   $0.748_{\pm0.055}$ | $0.786_{\pm0.080}$ | 5.08 |
| D: 64 / do: 0.0   $0.748_{\pm0.055}$ | $0.799_{\pm0.095}$ | 6.82 |
| D: 64 / do: 0.25   $0.748_{\pm0.055}$ | $0.798_{\pm0.095}$ | 6.68 |
| D: 64 / do: 0.5   $0.748_{\pm0.055}$ | $0.799_{\pm0.095}$ | 6.82 |
| D: 256 / do: 0.0   $0.748_{\pm0.055}$ | $\mathbf{0.800}_{\pm0.094}$ | 6.95 |
| D: 256 / do: 0.25   $0.748_{\pm0.055}$ | $0.799_{\pm0.094}$ | 6.82 |
| D: 256 / do: 0.5   $0.748_{\pm0.055}$ | $0.799_{\pm0.095}$ | 6.82 |

## A.10 EDGE HOMOPHILY IMPROVEMENT OF GRASS

In this section, we further explore how the performance enhancement is achieved by initializing current graph-based imputation methods with GRASS. Specifically, using GRASS as an initializer provides a warmed-up matrix along with a refined adjacency matrix. We assessed this adjacency matrix($\hat{\mathbf{A}}$) in terms of the homophily ratio, a crucial inductive bias in the graph domain. As shown in Table A.10, employing GRASS leads to an improved edge homophily ratio, indicating an increase in edges connecting nodes with the same labels. This improvement is attributed to the incorporation of feature gradients, which introduce task-relevant information through supervision signals from training the MLP. Consequently, a more refined graph structure, enriched with task-relevant information, allows current graph-based imputation methods to further enhance their performance by smoothing the representation of their nearest neighbors. In summary, this improvement in the graph structure is a key factor contributing to the performance boost observed with GRASS.

Table 19: Edge homophily ratio comparison between original adjacency matrix with refined adjacency matrix obtained via GRASS. Edge homophily ratio: $\frac{\text{number of edges connecting two nodes with same labels}}{\text{number of total edges}}$

| | **A** | **Â** | Impr. (%) |
|---|---|---|---|
| Mouse ES | 0.8591 | 0.9770 | 13.724 |
| Pancreas | 0.9319 | 0.9579 | 2.790 |
| Baron Human | 0.9557 | 0.9696 | 1.454 |
| Mouse Bladder | 0.5672 | 0.7559 | 33.269 |
| Breast Cancer | 0.6698 | 0.6701 | 0.045 |
| Hepatitis | 0.7902 | 0.8035 | 1.683 |
| Duke Breast | 0.6887 | 0.6915 | 0.407 |
| ADNI | 0.7130 | 0.7336 | 2.889 |
| ABIDE | 0.9142 | 0.9166 | 0.263 |

