# OpenReview forum: "Toward Generalizability of Graph-based Imputation on Bio-Medical Missing Data"
_ICLR.cc/2024/Conference — Submitted to ICLR 2024_

### Official Review · Reviewer_4SJf · 2023-10-30

**Soundness:** 1 poor
**Presentation:** 1 poor
**Contribution:** 2 fair
**Rating:** 3
**Confidence:** 3

**Summary:**

The paper proposes GRASS, a method to impute missing values in the features for tabular data using a graph-based method in the process. I could not fully understand how the method work, but, something like adding gradient feature to the current features, generate graph then using GNN based feature imputation.

**Strengths:**

The paper tinkers with different available ideas and it seems to have a good performance.

**Weaknesses:**

I'm struggling to find the idea of the method and how it differs from previous work.

First, the paper praised the graph-based imputation methods of how resilient they are against large missing rate in comparison to tabular data's methods  Mean, GAIN. This does not show the merit of graph-based methods since... they have additional graph information that tabular data methods do not have. This only show that the additional graph does complement the feature data, which is not at all a surprise. By generating a graph from tabular data, GRASS does not necessarily inherit the merit of these graph-based method that have
additional information in form of a graph.

Second, the paper shows the problem of deploying methods for filling in missing tabular data of "using incomplete feature" while it is not known how GRASS avoid the incomplete features.

Last but not least, I don't know how one trains a MLP model to get the gradient features from missing data.

**Questions:**

How could the method train a MLP model with missing values in the feature matrix?

---

> ### Author Response · Authors · 2023-11-22
> **Response to Reviewer 4SJf (1/2)**
>
> We sincerely appreciate your insightful feedback and recognition of the contributions made in our paper. We are committed to addressing your concerns comprehensively.
>
> **[Cons 1: Distinction from Previous Work]**
> The major difference of GRASS from other work is the generation of the graph information when the graph structure is not readily available and there are missing input features such as in the biomedical missing data scenario. We thus generate the graph information by utilizing the feature gradient obtained from the given tabular data, which does not require any external information. Compared to existing graph-based imputation methods, GRASS newly leverages the feature gradient to construct the graph structure by supplementing the noisy missing features. As Figure 3 demonstrates, the existing graph-based methods suffer from the low quality of the graph structure, and thus they occasionally underperform even basic methods like Mean Imputation where their graph construction depends solely on the noisy missing features. Therefore, GRASS has a clear contribution to better handle the challenging and realistic scenario (i.e., no given graph structure and missing input features) than the existing work that does not consider the characteristics of the scenario.
>
>
> **[Cons 2: Inheriting the Merit of Graph-based Methods in GRASS]**
> As pointed out, GRASS may not fully benefit from graph-based methods when the initial graph structure is absent. However, as Figure 1 illustrates, when the graph structure is well-defined, graph-based imputation methods can fully utilize the message-passing scheme (this would belong to the upper bound case of GRASS), an advantage not present in traditional tabular-based methods. In cases where the graph structure is not predefined, such as in the biomedical domain, the effectiveness of this message-passing scheme is uncertain. Our work investigates whether graph-based imputation can reach its full potential with a better-provided feature matrix and relevant graph structure. Figure 3 shows that when the graph structure is derived solely from the original missing feature matrix, it underperforms compared to classic tabular-based methods. However, utilizing GRASS as an initializer allows graph-based methods to surpass tabular-based methods by effectively leveraging the warmed-up matrix and adjacency matrix. This supports our research question: "Can graph-based imputation methods inherit the merit of the message-passing scheme with better-provided feature matrices and adjacency matrices?"
>
> **[Cons 3: Addressing Incomplete Features with GRASS]**
> Our approach specifically addresses scenarios with initially missing features. The term 'using incomplete features' might not fully align with our methodology. Rather than directly using incomplete features, we opt for task-relevant supplements to build the graph structure. As highlighted in the introduction, generating a graph from incomplete features is challenging and often leads to suboptimal results. This challenge motivates our approach to obtain a feature gradient by training an MLP layer, requiring only the input feature matrix. This gradient plays a crucial role as a supplement in constructing a column-wise graph. Thus, GRASS avoids using incomplete features directly and instead equips the model with task-relevant feature gradients for imputation via the generated column-wise graph.

---

> > ### Author Response · Authors · 2023-11-22
> > **Response to Reviewer 4SJf (2/2)**
> >
> > **[Question 1: Training an MLP Model with Missing Values]**
> > As mentioned in the footnote on page 4 of our paper, we employed zero imputation for initially missing values during MLP training. More precisely, the input for the MLP consists of the zero-imputed original feature matrix ($\mathbf{X}$). By conducting a supervised downstream task (e.g., classification task), we can obtain the feature gradient from the MLP . This aligns with our primary objective for training the MLP, which is to obtain feature gradients that contain task-relevant information. This information is vital when generating a column-wise graph and distinguishes our work from existing methods, where graph construction does not incorporate such task-relevant information. For additional details regarding the process of obtaining feature gradients, we have included a comprehensive explanation of the implementation perspective. We respectfully request the reviewer to refer to the information provided in Listing 1 on page 21 for further insights.

---

> ### Author Response · Authors · 2023-11-23
> **Gentle Reminder for Reviewer 4SJf**
>
> This is a gentle last-minute reminder regarding the rebuttal period. We have diligently addressed the concerns raised by the reviewers, as outlined above, and are hopeful for a positive response in the final stage of evaluating our work. We would like to highlight that the aforementioned better-graph structure in **[Cons 2: Inheriting the Merit of Graph-based Methods in GRASS]** can be evaluated by the below experiments.
>
> | Edge Homophily |   $\mathbf{A}$   |  $\hat{\mathbf{A}}$ | Impr. (%) |
> |:--------------:|:------:|:------:|:---------:|
> |    Mouse ES    | 0.8591 | 0.9770 |   13.724  |
> |    Pancreas    | 0.9319 | 0.9579 |   2.790   |
> |   Baron Human  | 0.9557 | 0.9696 |   1.454   |
> |  Mouse Bladder | 0.5672 | 0.7559 |   33.269  |
> |  Breast Cancer | 0.6698 | 0.6701 |   0.045   |
> |    Hepatitis   | 0.7902 | 0.8035 |   1.683   |
> |   Duke Breast  | 0.6887 | 0.6915 |   0.407   |
> |      ADNI      | 0.7130 | 0.7336 |   2.889   |
> |      ABIDE     | 0.9142 | 0.9166 |   0.263   |
>
>
> We have also included this experiment in Appendix A.10, on Page 26.
>
> As shown in the above table, employing GRASS to obtain refined adjacency matrix ($\hat{\mathbf{A}}$) leads to an improved edge homophily ratio, indicating an increase in edges connecting nodes with the same labels. This improvement is attributed to the incorporation of feature gradients, which introduce task-relevant information through supervision signals from training the MLP. Consequently, a more refined graph structure, enriched with task-relevant information, allows current graph-based imputation methods to further enhance their performance by smoothing the representation of their nearest neighbors. In summary, this improvement in the graph structure is a key factor contributing to the performance boost observed with GRASS.
>
> Overall, we hope that these experimental results will provide the reviewer with additional clarity and be taken into consideration while making the final decision on our work.
> Thank you for your invaluable time.
>
> Best,
> Authors

---

### Official Review · Reviewer_T1VM · 2023-10-31

**Soundness:** 3 good
**Presentation:** 3 good
**Contribution:** 3 good
**Rating:** 6
**Confidence:** 5

**Summary:**

This paper tackles the issue of missing feature imputation on bio-medical tabular data. The authors adapt techniques originally developed for graphs with missing features to tabular data by introducing a kNN adjacency matrix. They utilize feature gradient as additional features to construct a column-wise adjacency matrix using kNN. They perform colum-wise feature propagation with the matrix and prepare a row-wise adjacency matrix. This row-wise adjacency matrix is employed in graph-based imputation methods. They evaluate their approach using medical and bio-domain datasets.

**Strengths:**

1. The idea is easy to understand.
2. The paper is well written.

**Weaknesses:**

$\textbf{1. Not novel}$

The authors overclaim that “We, for the first time, explore the generalizability of recent graph-based imputation models in the context of real-world tabular data with missing values.” It is not true. scFP [1] utilizes FP, a recent graph-based imputation model, for bio-medical tabular data with missing values. scFP also constructs a kNN graph for tabular data given without an adjacency matrix, like this paper. The proposed method in this paper is built on scFP and experimental setting in this paper refers to [1]. However, scFP is not compared with the proposed method in experiments. Furthermore, all the components in this paper are described as they are novel. To claim the effectiveness of new components proposed in this paper, it is essential to explain and cite components from other work and demonstrate the performance gain over scFP. Moreover, constructing a kNN graph to use graph-based methods on tabular data has been extensively explored [2, 3, 4, 5] and should be credited.

 $\textbf{2. Issues in Experiments}$

2-1. Unfair hyper-parameter setting

Regarding the experiments, I disagree that using the same hyper-parameters is fair. The authors state, “We set a consistent dropout rate of 0.5 and a hidden dimension of 64 across all methods.” This implies that a dropout rate of 0.5 and a hidden dimension of 64 are universally applied, irrespective of the datasets. However, all machine learning methods have their own optimal hyper-parameters for a given setting (e.g., dataset, domain, the type of task). For example, GCN [6] set sets of hyper-parameters as follows:
- Citeseer, Cora, and Pubmed: 0.5 (dropout rate), 16 (hidden dimension)
- NELL: 0.1 (dropout rate), 64 (hidden dimension)
Moreover, it is not fair to use the same hyper-parameters for downstream neural networks across all methods, not only in terms of target datasets but also in terms of methods. For example, PaGNN [7] set the hidden dimension to 16 on Cora and Citeseer while it set one to 64 on AmaPhoto and AmaComp. In the case of FP [8] and PCFI [9], they set the hidden dimension to 256 for OGBN-Arxiv.

When applying methods validated in citation and recommendation networks to unfamiliar domains, careful consideration is needed for hyperparameter settings. In [10], a study comparing GNNs and traditional methods for molecular property prediction demonstrates reasonable search spaces for each hyper-parameter across all methods, including GNNs and others.

2-2. Omitting an important baseline

The authors mention scFP [1] in the related work and experimental details sections while they exclude scFP in experiments. scFP deals with medical/bio datasets used in this paper and also utilizes kNN graphs before FP, which is very similar to this paper.

2-3. Downstream neural  networks

The authors state, “To align with our focus on downstream tasks, we appended a logistic classifier to methods that exclusively target imputation”. Does it mean the authors commonly use logistic classifiers for all imputation methods, including FP, PCFI, GAIN, GRAPE, etc.? If so, FP and PCFI do not use downstream neural networks where message passing occurs, unlike  GCNMF.

2-4. No error bar

The authors state, “For the dataset split, we randomly generated five different splits with train/val/rest ratio in 10%, 10%, 80%.” Are all the reported values from those five splits? It needs to be clarified. Then there is no error bar. The performance differences between the methods are very small. For all the datasets, the performance gain of the best performing method over the second best one is very marginal. Therefore, it seems necessary to check the error bars.

$\textbf{3. Convergence issue}$

The authors state, “Given approach uses a custom kNN graph, discussions about convergence can be found in Appendix A.2.” However, the authors’ message through the claim in Appendix A.2 is not clear. After all, kNN graph with k between 1 and 10 is not a strongly connected graph in general, which does not satisfy the convergence condition of FP. It means the warmed-up matrix does not converge even after large enough K.

$\textbf{4. Initialization issue}$

There is no explanation for $X^{0}_{v,d}$ in Eq (2). Do the authors fill in missing values with zeros? This needs clarification. However, as convergence is not guaranteed, the output depends on the initial values. Hence justifying zero imputation for the initially missing values is questionable if it is employed.

$\textbf{5. Ineffectiveness of the clamping process}$

The clamping process designed for categorical features can be applied only for the datasets (Breast Cancer, Hepatitis, Duke Breast, ADNI, and ABIDE) with categorical features. However, the gain obtained with GRASS on the five datasets is much smaller than that in the other datasets. Thus the clamping process seems to contribute very little to the performance improvement.

$\textbf{6. No ablation study}$

Since where the performance gain comes from is not clear, an ablation study should be performed. Especially column-wise feature propagation without feature gradients should be compared with GRASS.

$\textbf{7. To claim contribution in exploring generalizability}$

To claim contribution in exploring generalizability of recent graph-based imputation models in the real-world tabular data with missing values, it seems necessary to demonstrate the performance not on datasets within a single domain but on datasets used for the comparison of existing tabular data imputation methods. For example, Concrete, Housing, Wine, and Yacht from the UCI Machine Learning repository [11].

$\textbf{Minor comments/suggestions}$

* The colors of the bars in Figure 3 are too similar. It might be better to consider an alternative visualization.
* According to the source code, $A^{feat}$ and $\hat{A}$ are weighted adjacency matrices where weights are cosine similarity. However, based on the manuscript alone, it seems that $A^{feat}$ and $\hat{A}$ are unweighted adjacency matrices are generated through kNN while $\tilde{A}$ is a weighted adjacency matrix obtained via normalization. Therefore, it is necessary to explicitly specify that $A^{feat}$ and $\hat{A}$ are weighted by cosine similarity.
* calmping categorical columns -> clamping categorical columns (page 18)
* legnth -> length (page 18)

[1] Single-cell rna-seq data imputation using feature propagation
[2] EGG-GAE: scalable graph neural networks for tabular data imputation
[3] Fusion of Graph and Tabular Deep Learning Models for Predicting Chronic Kidney Disease
[4] Leveraging graph convolutional networks for semi-supervised fault diagnosis of HVAC systems in data-scarce contexts
[5] TabGSL: Graph Structure Learning for Tabular Data Prediction
[6] Semi-Supervised Classification with Graph Convolutional Networks
[7] Incomplete Graph Representation and Learning via Partial Graph Neural Networks
[8] On the unreasonable effectiveness of feature propagation in learning on graphs with missing node features
[9] Confidence-based feature imputation for graphs with partially known features.
[10] Could graph neural networks learn better molecular representation for drug discovery?
[11] UCI Machine Learning repository

**Questions:**

Q1. Could the authors provide results on Concrete, Housing, Wine, and Yacht from various domains?

Q2. Could the authors provide full results table including standard deviation?

Q3. How GRASS improves the performance of graph-structure-agnostic methods (e.g., GAIN and GRAPE) ?

---

> ### Author Response · Authors · 2023-11-22
> **Response to Reviewer T1VM (1/8)**
>
> We deeply appreciate the reviewer's dedicated time in reviewing our paper and providing valuable, thorough feedback, which has significantly contributed to the improvement of our work. We have done our utmost during the rebuttal period to address the concerns raised, and we now respectfully offer a point-by-point response to the reviewer’s valuable insights.
>
> **[Cons 1. Novelty regarding scFP]**
>
> The reviewer's concerns about novelty in comparison to scFP are acknowledged. We respectfully argue that our work, GRASS, is distinct from scFP in the following three key areas: Target Domain, Target Task and Imputation Methodology, and Usage of Feature Propagation. Here are the details for each:
>
> ***(1) Target Domain***: While scFP focuses on the single-cell RNA-seq (scRNA-seq) domain, particularly from a biological perspective, GRASS adopts a more general approach for the broader `bio-medical' domain, as indicated in the paper's title. This distinction is crucial as scRNA-seq datasets typically comprise numerical features where each element represents the count of a gene's RNA transcript sequenced by the sequencing machine. In contrast, medical datasets often include both numerical and categorical features, such as patient information. This versatility underscores the broader applicability of GRASS, capable of handling both numerical and categorical features, the latter through the clamping technique as discussed in Section 3.3. Therefore, we argue that the target domain of scFP, primarily focused on numerical matrix imputation in scRNA-seq, differs from that of GRASS, which extends to handling categorical data often encountered in patient data.
>
> ***(2) Target Task and Imputation Methodology***: Unlike scFP, which is unsupervised with its primary goal being effective imputation in sparse and noisy cell-gene count matrices, this work concentrates on supervised tasks, specifically on downstream applications like classification. Notably, the objective of imputation is often to enhance performance in relevant downstream tasks [1, 2]. In this context, while the unsupervised approach of scFP can align with supervised tasks through probing (i.e., attaching a classifier), it is important to note that since its imputation occurs prior to probing, scFP cannot incorporate any downstream task-related knowledge during the imputation process, potentially leading to shortcomings in classification tasks. Conversely, as GRASS is directly designed with downstream tasks in mind, it incorporates knowledge pertinent to these tasks during imputation. This is achieved by utilizing the feature gradient, which is obtained during training 2-layer MLP. This fundamental difference in the target task (classification vs. imputation) and the imputation process (incorporating relevant downstream knowledge or not) distinctly sets the two methodologies apart.
>
> ***(3) Usage of Feature Propagation***: Although both scFP and GRASS employ Feature Propagation (FP), their applications of this process differ significantly. Specifically, scFP utilizes FP from a row-wise perspective, i.e., focusing on cell-cell relationships while assuming gene-gene relationship independence. Although beneficial for smoothing similar and relevant samples, this approach does not capture interactions between columns (features), which are pivotal in the biomedical domain. For instance, in scRNA-seq, gene-gene relationships, such as co-expression networks, play a critical role in identifying key regulatory genes or pathways, offering insights into underlying biological or disease mechanisms [3,4]. Acknowledging this, GRASS initially employs column-wise FP to capture potential feature interactions, e.g., gene-gene relationships. It's also noteworthy that GRASS incorporates not only the feature matrix but also the feature gradient relevant to downstream tasks when generating the column-wise $k$NN graph. Consequently, before initiating row-wise (sample-wise) smoothing in the relevant GNN model, GRASS is able to consider feature relationships that scFP does not capture. This distinction is illustrated in Figure 8 in Page 23  and significantly differentiates the two methodologies.
>
> (link to Figure 8: https://anonymous.4open.science/r/grass-iclr-41D5/figs/crop_scfp_grass_2.pdf)
>
> [1] Rossi, Emanuele, et al. "On the unreasonable effectiveness of feature propagation in learning on graphs with missing node features." Learning on Graphs Conference. PMLR, 2022.
> [2] Wang, Juexin, et al. "scGNN is a novel graph neural network framework for single-cell RNA-Seq analyses." Nature communications 12.1 (2021): 1882.
> [3] Cochain, Clément, et al. "Single-cell RNA-seq reveals the transcriptional landscape and heterogeneity of aortic macrophages in murine atherosclerosis." Circulation research 122.12 (2018): 1661-1674.
> [4] Galfre, Silvia Giulia, et al. "COTAN: scRNA-seq data analysis based on gene co-expression." NAR Genomics and Bioinformatics 3.3 (2021): lqab072.

---

> ### Author Response · Authors · 2023-11-22
> **Response to Reviewer T1VM (2/8)**
>
> **[Cons 2: Issues in Experimental Settings - Hyper-parameters, Baseline, Downstream Neural Network, Error Bar]**
>
> **2-1. Hyper-parameters**: We would like to respectfully assert that our hyper-parameter setup is fair and appropriate. Considering that our work is largely based on Feature Propagation, which in the original paper utilized a structure of 2 layers with a hidden dimension of 64 and a dropout rate of 0.5, our approach aligns with established practices. In terms of the downstream GNN model (GCN), which is widely used for graph-based imputation, adding complexity to this model could unnecessarily complicate the comparison of imputation methods. A fixed hyper-parameter setting for the downstream network acts as a control factor, allowing us to focus on the experimental factor, which is the method of imputation, such as GCN-Zero, GCN-NM, or GCN-FP. This approach enables a direct comparison of the impact of the imputation module.
>
> Furthermore, we would like to note that the datasets mentioned by the reviewer, namely Cora, CiteSeer, and PubMed, are not within the scope of our study, as we focus on tabular data rather than graph-structured data. Therefore, the more relevant hyperparameters in our context pertain to the imputation step, particularly the number of connected neighbors. To this end, we have allowed a flexible search space for all graph-based models, with options among {1,3,5,10}.
>
> Additionally, in the computational biology domain, which is our primary focus, there is a preference for a unified hyper-parameter configuration across different datasets to ensure reproducibility and generalizability across various types of biological data. Nonetheless, we acknowledge the importance placed on hyper-parameter tuning in the broader ML community. With this in mind, we will ensure a more thorough examination of hyper-parameters in our final version of the paper.
>
>
> **2-2. Baseline**: As pointed out by the reviewer, we initially omitted scFP in our submission, focusing more on general approaches prevalent in graph-based imputation methods and conventional methodologies within the realm of tabular data. It is important to note, based on our understanding, that scFP is tailored specifically for the single-cell RNA-seq (scRNA-seq) domain and primarily evaluated on bio datasets, not covering medical datasets. However, recognizing the similarities in process with scFP, and in response to the reviewer's request, we have expanded our evaluation to include scFP alongside our current baselines across all datasets. This broader evaluation approach allows us to provide a more comprehensive comparison, particularly in bio datasets, where scFP's performance can be directly contrasted with other methods. The results of this extended evaluation, particularly focusing on bio datasets, are presented below. We can observe that GRASS-initialized graph-based imputation methods consistently outperform scFP. Moreover, interestingly, scFP with GRASS initialization enhances the performance up to **16.23**%, which implies that GRASS can provide distinct knowledge from that of scFP, i.e., column-wise relationship.

---

> ### Author Response · Authors · 2023-11-22
> **Response to Reviewer T1VM (3/8)**
>
> |          | Mouse ES (IMR: 27.21%) |              |           | Pancreas (IMR: 56.65%) |             |           | Baron Human (IMR: 57.25%) |             |           | Mouse Bladder (IMR: 69.05%) |             |           |
> |----------|:----------------------:|:------------:|:---------:|:----------------------:|:-----------:|:---------:|:-------------------------:|:-----------:|:---------:|:---------------------------:|:-----------:|:---------:|
> |          |           OG           |  GRASS init. | Impr. (%) |           OG           | GRASS init. | Impr. (%) |             OG            | GRASS init. | Impr. (%) |              OG             | GRASS init. | Impr. (%) |
> | LP       |       0.878+0.005      | 0.9794+0.003 |   11.43   |       0.656+0.039      | 0.798+0.068 |   21.66   |        0.736+0.022        | 0.828+0.055 |   12.46   |         0.556+0.030         | 0.643+0.053 |   15.57   |
> | GCNMF    |       0.525+0.238      | 0.9729+0.008 |   85.31   |       0.527+0.210      | 0.708+0.087 |   34.27   |        0.350+0.130        | 0.817+0.066 |   133.30  |         0.300+0.182         | 0.701+0.042 |   133.90  |
> | PaGNN    |       0.899+0.072      | 0.9806+0.002 |    9.03   |       0.701+0.044      | 0.768+0.040 |    9.58   |        0.777+0.043        | 0.820+0.057 |    5.53   |         0.713+0.056         | 0.775+0.028 |    8.78   |
> | GCN-zero |       0.960+0.005      | 0.9823+0.004 |    2.30   |       0.687+0.066      | 0.783+0.062 |   14.02   |        0.812+0.030        | 0.842+0.049 |    3.71   |         0.712+0.015         | 0.768+0.031 |    7.83   |
> | GCN-nm   |       0.885+0.098      | 0.9823+0.004 |   10.99   |       0.679+0.047      | 0.788+0.068 |   16.09   |        0.758+0.045        | 0.801+0.084 |    5.71   |         0.721+0.050         | 0.775+0.030 |    7.38   |
> | GCN-FP   |       0.900+0.100      | 0.9825+0.005 |    9.17   |       0.716+0.046      | 0.788+0.068 |   10.08   |        0.789+0.039        | 0.802+0.084 |    1.61   |         0.686+0.048         | 0.772+0.036 |   12.48   |
> | PCFI     |       0.949+0.004      |  0.955+0.006 |    0.57   |       0.673+0.055      |  0.686+0.04 |    1.95   |        0.769+0.036        | 0.792+0.038 |    2.96   |         0.710+0.046         | 0.727+0.028 |    2.41   |
> | Mean     |       0.979+0.006      | 0.9799+0.004 |    0.08   |       0.616+0.044      | 0.619+0.032 |    0.44   |        0.672+0.010        | 0.694+0.023 |    3.41   |         0.555+0.074         | 0.569+0.062 |    2.39   |
> | kNN      |       0.969+0.011      |  0.977+0.005 |    0.83   |       0.652+0.047      | 0.706+0.048 |    8.34   |        0.746+0.048        | 0.760+0.053 |    1.82   |         0.587+0.038         | 0.674+0.059 |   14.82   |
> | GAIN     |       0.978+0.011      | 0.9826+0.007 |    0.39   |       0.638+0.075      | 0.738+0.024 |   15.66   |        0.728+0.041        | 0.745+0.033 |    2.39   |         0.585+0.030         | 0.649+0.038 |   10.84   |
> | MIWAE    |           OOM          |       -      |     -     |            OOM           |      -      |     -     |             OOM             |      -      |     -     |              OOM              |      -      |     -     |
> | GRAPE    |           OOM          |       -      |     -     |            OOM           |      -      |     -     |             OOM             |      -      |     -     |              OOM              |      -      |     -     |
> | IGRM     |           OOM          |       -      |     -     |            OOM           |      -      |     -     |             OOM             |      -      |     -     |              OOM              |      -      |     -     |
> | scFP     |       0.952+0.004      |  0.976+0.003 |    2.52   |       0.743+0.044      | 0.788+0.085 |    6.05  |        0.809+0.067        | 0.853+0.031 |    5.43   |         0.653+0.024         | 0.759+0.022 |    16.23   |

---

> ### Author Response · Authors · 2023-11-22
> **Response to Reviewer T1VM (4/8)**
>
> For Medical datasets the overall performance is as below:
>
> |          | Breast Cancer (IMR: 0.35%) |             |           | Hepatitis (IMR: 5.67%) |             |           | Duke Breast (IMR: 11.94%) |             |           | ADNI (IMR: 30.02%) |             |           | ABIDE (IMR: 69.74%) |             |           |
> |----------|:--------------------------:|:-----------:|:---------:|:----------------------:|:-----------:|:---------:|:-------------------------:|:-----------:|:---------:|:------------------:|:-----------:|:---------:|:-------------------:|:-----------:|:---------:|
> |          |             OG             | GRASS init. | Impr. (%) |           OG           | GRASS init. | Impr. (%) |             OG            | GRASS init. | Impr. (%) |         OG         | GRASS init. | Impr. (%) |          OG         | GRASS init. | Impr. (%) |
> | LP       |         0.561+0.038        | 0.562+0.041 |    0.14   |       0.573+0.078      | 0.608+0.053 |    6.06   |        0.672+0.021        | 0.678+0.026 |    0.98   |     0.928+0.005    | 0.943+0.005 |    1.56   |     0.894+0.009     | 0.895+0.011 |    0.13   |
> | GCNMF    |         0.551+0.033        | 0.579+0.049 |    5.02   |       0.685+0.097      | 0.707+0.088 |    3.22   |        0.664+0.035        | 0.688+0.032 |    3.61   |     0.897+0.045    | 0.944+0.004 |    5.25   |     0.819+0.042     | 0.913+0.010 |   11.49   |
> | PaGNN    |         0.540+0.037        | 0.562+0.032 |    3.98   |       0.729+0.074      | 0.741+0.058 |    1.74   |        0.685+0.033        | 0.690+0.029 |    0.69   |     0.953+0.003    | 0.955+0.003 |    0.27   |     0.907+0.009     | 0.914+0.008 |    0.82   |
> | GCN-zero |         0.542+0.048        | 0.557+0.039 |    2.71   |       0.713+0.090      | 0.714+0.088 |    0.14   |        0.673+0.022        | 0.694+0.021 |    3.13   |     0.956+0.003    | 0.957+0.003 |    0.17   |     0.902+0.008     | 0.915+0.008 |    1.38   |
> | GCN-nm   |         0.538+0.049        | 0.566+0.052 |    5.07   |       0.702+0.071      | 0.702+0.071 |    0.00   |        0.678+0.033        | 0.691+0.025 |    1.96   |     0.955+0.003    | 0.956+0.003 |    0.19   |     0.905+0.011     | 0.918+0.007 |    1.48   |
> | GCN-FP   |         0.543+0.047        | 0.565+0.052 |    4.05   |       0.705+0.085      | 0.707+0.092 |    0.20   |        0.661+0.031        | 0.688+0.028 |    4.21   |     0.955+0.003    | 0.957+0.003 |    0.18   |     0.908+0.014     | 0.915+0.005 |    0.86   |
> | PCFI     |         0.545+0.039        | 0.547+0.040 |    0.44   |       0.728+0.108      | 0.728+0.108 |    0.00   |        0.693+0.029        | 0.696+0.030 |    0.40   |     0.951+0.004    | 0.955+0.003 |    0.46   |     0.915+0.008     | 0.917+0.010 |    0.26   |
> | Mean     |         0.562+0.045        | 0.562+0.045 |    0.00   |       0.691+0.072      | 0.711+0.081 |    2.86   |        0.687+0.018        | 0.687+0.019 |    0.04   |     0.939+0.002    | 0.943+0.003 |    0.46   |     0.607+0.027     | 0.905+0.007 |   49.09   |
> | kNN      |         0.552+0.041        | 0.556+0.041 |    0.67   |       0.612+0.097      | 0.626+0.105 |    2.15   |        0.692+0.026        | 0.697+0.014 |    0.74   |     0.943+0.003    | 0.943+0.004 |    0.01   |     0.896+0.009     | 0.907+0.010 |    1.16   |
> | GAIN     |         0.566+0.044        | 0.567+0.043 |    0.21   |       0.578+0.093      | 0.646+0.080 |   11.63   |        0.699+0.018        | 0.699+0.017 |    0.09   |     0.937+0.003    | 0.944+0.003 |    0.67   |     0.793+0.010     | 0.910+0.009 |   14.70   |
> | MIWAE    |         0.558+0.033        | 0.563+0.035 |    0.93   |       0.573+0.080      | 0.608+0.077 |    6.25   |        0.692+0.013        | 0.693+0.012 |    0.13   |          OOM         |      -      |     -     |     0.623+0.015     | 0.898+0.008 |   44.10   |
> | GRAPE    |         0.572+0.029        | 0.573+0.017 |    0.26   |       0.701+0.033      | 0.706+0.032 |    0.63   |            OOM            |      -      |     -     |          OOM         |      -      |     -     |     0.889+0.010     | 0.906+0.006 |    1.90   |
> | IGRM     |         0.548+0.039        | 0.552+0.037 |    0.66   |       0.668+0.087      | 0.703+0.109 |    5.26   |            OOM            |      -      |     -     |          OOM         |      -      |     -     |     0.747+0.019     | 0.908+0.004 |   21.54   |
> | scFP     |         0.554+0.047        | 0.563+0.055 |    1.62   |       0.691+0.077      | 0.691+0.077 |    0.00   |        0.678+0.031        | 0.690+0.030 |    1.76   |     0.953+0.003    | 0.954+0.002 |    0.10   |     0.894+0.010     | 0.903+0.007 |    1.00   |
>
> We have taken steps to enhance the visibility of the tables discussed in our responses. These tables have been presented in a more accessible format in Appendix A.5 on Page 22 of our manuscript. We kindly ask the reviewers to refer to this Appendix for a detailed view of the results.

---

> ### Author Response · Authors · 2023-11-22
> **Response to Reviewer T1VM (5/8)**
>
> **2-3. Downstream Neural Network**: As the reviewer correctly noted, we have indeed appended a logistic classifier to methods primarily focused on imputation. For graph-based imputation methods like GCNMF, PaGNN, FP, and PFCI, which target downstream tasks rather than solely imputation, we utilized a standard 2-layer GCN. Conversely, for methods such as Mean, GAIN, kNN, GRAPE, and IGRM, which are more imputation-centric, we employed a logistic classifier (1-layer MLP) for downstream tasks. Notably, while GRAPE and IGRM incorporate a graph structure, our empirical findings indicated that using a logistic classifier post-imputation yielded better results than directly employing their GNN for downstream tasks. This was primarily due to the differing loss functions used in imputation (MSE loss) and classification (CE loss). Therefore, we decided to consistently use a logistic classifier for models where the primary focus is on imputation.
>
> **2-4. Error Bar**: We appreciate the reviewer pointing out the need for clearer error bar notation. We conducted our experiments with five random splits, reporting the average scores. To provide a clearer picture of the variability in our results, we have now included the standard deviation alongside the average scores in the tables presented in the main text and Appendix A.5 on Page 22. Additionally, to highlight the best-performing and most-improved models, we have used bold formatting for the best model and underlining for the most improved model in these tables.

---

> ### Author Response · Authors · 2023-11-22
> **Response to Reviewer T1VM (6/8)**
>
> **[Cons 3: Convergence Issue]**
> We greatly value the reviewer's expertise in feature propagation and related works concerning missing features, and we are thankful for bringing this crucial discussion to the forefront. We concur that the convergence property is of paramount importance in the missing feature community, and we have delved deeper to articulate our perspective on this issue. As stated in our initial submission, we acknowledge that strong connectivity in kNN graph generation is not always guaranteed, and achieving convergence may not always be advantageous.
>
> To examine whether the convergence property contributes to performance improvement, we conducted an empirical analysis. In the table below, we extended the range of $ k_\text{col} $ and $ k_\text{row} $ values from \{1,3,5,10\} to as high as 50, to test the connectivity of the resulting graphs. We observed that when $ k_\text{col} $ and $ k_\text{row} $ exceed 10, the graph becomes strongly connected, implying that every node is reachable from every other node. Interestingly, while strong connectivity facilitates the selection of neighbor numbers and meets the necessary condition for FP to converge, it does not automatically lead to performance improvements. Optimal performance was actually achieved within the initially proposed smaller range of $ k $ values. Further analysis revealed that increasing $ k_\text{col} $ resulted in an oversmoothing issue in the output, particularly noticeable in the warmed-up matrix. This effect was quantified using the MADGap metric [1], which assesses the representational differences between remote and neighboring nodes. Our findings suggest that in bio-medical tabular data, where an initial graph structure is not provided and a $ k $NN graph must be manually constructed, opting for a larger $ k $ value to exploit the convergence property of FP may not be the most effective strategy, especially in scenarios with significant missing data.
>
> | Mouse ES (Macro-F1) |    | k_row |       |       |       |   |          |
> |:-------------------:|:--:|:-----:|:-----:|:-----:|:-----:|---|----------|
> |                     |    | 1     | 5     | 10    | 50    |   | MADGap   |
> |        K_col        | 50  | 0.893 | 0.840 | 0.841 | 0.842 |   | 1.43E-06  |
> |                     | 10 | 0.910 | 0.983 | 0.923 | 0.950 |   |  0.00085 |
> |                     | 5 | 0.767 | 0.919 | 0.977 | 0.945 |   | 0.00341  |
> |                     | 1 | 0.836 | 0.962 | 0.964 | 0.946 |   | 0.00355 |
>
> | ABIDE (AUROC) |    | k_row |       |       |       |   |          |
> |:-------------:|:--:|:-----:|:-----:|:-----:|:-----:|---|----------|
> |               |    | 1     | 5     | 10    | 50    |   | MADGap   |
> |     K_col     | 50  | 0.905 | 0.908 | 0.906 | 0.902 |   | 0.00E+00  |
> |               | 10  | 0.889 | 0.881 | 0.888 | 0.895 |   | 0.00148  |
> |               | 5 | 0.894 | 0.894 | 0.901 | 0.904 |   |  0.01134 |
> |               | 1 | 0.911 | 0.904 | 0.910 | 0.905 |   | 0.01048 |
>
>
> We have provided more visual illustrations in Figure 7, Appendix A.2, on Page 16.
>
> (link to Figure 7: https://anonymous.4open.science/r/grass-iclr-41D5/figs/crop_discussion_k.pdf)
>
> To summarize, in the context of manually generated $ k $NN graphs, particularly in bio-medical datasets where an adjacency matrix is not inherently available, strong connectivity does not necessarily guarantee performance improvement. Such settings are prone to oversmoothing, and hence, convergence may not always be beneficial for classification tasks.
>
> [1] Chen, Deli, et al. "Measuring and relieving the over-smoothing problem for graph neural networks from the topological view." Proceedings of the AAAI conference on artificial intelligence. Vol. 34. No. 04. 2020.

---

> > ### Author Response · Authors · 2023-11-22
> > **Response to Reviewer T1VM (7/8)**
> >
> > **[Cons 4: Initialization Issue]**
> > Thank you for highlighting the initial value used in the initial state of the feature matrix. During the submission, we made footnote on Page 4 that we used zero imputation for the initial missing values. In cases where convergence is not guaranteed, the output indeed depends on the initial values for missing data. The rationale behind using zero imputation aligns with natural environments in the bio-medical domain. In the bio domain, for instance, scRNA-seq data often experiences dropout phenomena, leading to false zeros. Zero imputation helps flag these false zeros and differentiate them from non-zero values. Similarly, in the medical domain, datasets often consist of a mix of numerical and categorical values. After one-hot encoding categorical features, matrices are predominantly filled with zeros. Thus, starting with zero imputation is a natural and safe choice.
> >
> > **[Cons 5: Effectiveness of Clamping Process]**
> > The marginal performance gain in medical datasets, compared to bio datasets, is primarily due to the significantly smaller number of training samples in medical datasets, as shown in Table 2. We intentionally selected a 10/10/80 training/validation/test split to simulate a realistic and challenging training environment with limited training data. Therefore, the observed performance difference is not a limitation of the clamping process but rather a reflection of the available training samples.
> >
> > **[Cons 6: Ablation Study]**
> > We believe there might have been an oversight regarding our ablation study, which is included in Table 1 of the main paper. This study demonstrates the benefits of utilizing the feature gradient in conjunction with column-wise feature propagation, as opposed to using only the input feature matrix. Additionally, the effectiveness of the clamping technique is evident when comparing the results in rows 3, 4, and 5 of Table 1, addressing the concerns raised in **[Cons 5]**.
> >
> > **[Cons 7: Claim on Generalizability]**
> > As indicated in the paper's title, our use of 'generalizability' primarily refers to various datasets and scenarios within the bio-medical domain. We have clarified this in the updated PDF, specifying that our claims of generalizability are within this domain. While our method is broadly applicable, extending our investigation to domains such as product and social sciences falls outside the scope of this work. Nevertheless, to further address the reviewer's request, we have conducted additional experiments in these domains, as detailed in the following response to Question 1.
> >
> > **[Minor Comments]**
> > We thank the reviewer for pointing out minor comments and typos. In response, we have reformatted Figure 3 into a table format and addressed other minor comments, highlighting the changes in orange text in the updated PDF.

---

> > > ### Author Response · Authors · 2023-11-22
> > > **Response to Reviewer T1VM (8/8)**
> > >
> > > **[Question 1: Results on Concrete, Housing, Wine, Yacht Domains]**
> > > This inquiry aligns with [Cons 7: Claim on Generalizability], as discussed previously. While our primary focus is on the bio-medical domain, we acknowledge the potential to extend our methodology to the domains suggested by the reviewer. However, it's pertinent to note that datasets like Concrete, Housing, and Yacht are oriented towards 'regression' tasks, diverging from the 'classification' focus of our paper. Therefore, we present the results for the Wine dataset, which aligns more closely with our classification-centric approach.
> > >
> > > It is also crucial to mention that our initial research objective targets datasets with inherently missing data. Upon reviewing the original Wine dataset, we noted the absence of missing values, which precludes the necessity for imputation in its native form. To adapt to this scenario, we artificially introduced missing data by randomly dropping 30% of the feature matrix. This approach allowed us to conduct experiments under conditions that necessitate imputation, consistent with the focus of our research. The results of these experiments are presented below, demonstrating the application of our methodology to the Wine dataset within the framework of our study's primary objectives.
> > >
> > >
> > >
> > > |          | WINE   (IMR: 0.00%) |             |           |
> > > |----------|:-------------------:|:-----------:|:---------:|
> > > |          |          OG         | GRASS init. | Impr. (%) |
> > > | LP       |     0.647+0.017     | 0.647+0.017 |    0.00   |
> > > | GCNMF    |     0.656+0.027     | 0.657+0.023 |    0.11   |
> > > | PaGNN    |     0.650+0.030     | 0.661+0.023 |    1.65   |
> > > | GCN-zero |     0.637+0.043     | 0.648+0.033 |    1.68   |
> > > | GCN-nm   |     0.629+0.034     | 0.660+0.030 |    4.93   |
> > > | GCN-FP   |     0.642+0.032     | 0.647+0.042 |    0.79   |
> > > | PCFI     |     0.650+0.042     | 0.670+0.031 |    3.06   |
> > > | Mean     |     0.585+0.028     | 0.600+0.021 |    2.46   |
> > > | kNN      |     0.629+0.012     | 0.640+0.012 |    1.75   |
> > > | GAIN     |     0.618+0.012     | 0.640+0.018 |    3.61   |
> > > | MIWAE    |     0.514+0.027     | 0.591+0.024 |   14.93   |
> > > | GRAPE    |     0.567+0.064     | 0.587+0.045 |    3.52   |
> > > | IGRM     |     0.573+0.022     | 0.579+0.044 |    0.96   |
> > > | scFP     |     0.620+0.022     | 0.620+0.026 |    0.00   |
> > >
> > > We have observed that GRASS aligns seamlessly with existing baselines, enhancing the performance of the original models in relevant downstream tasks. This demonstrates its generalizability across different data types and settings.
> > >
> > > **[Question 2: Full Result Table Including Standard Deviation]**
> > > In response to the reviewer's request for a detailed results table, including standard deviation, we have provided an overview in the above section **[Cons 2]**. For a more comprehensive view, including the standard deviation of our results, we invite the reviewer to refer to Appendix A.5., on Page 22 of our manuscript.
> > >
> > > **[Question 3: How GRASS Improves Performance of Graph-Structure-Agnostic Methods]**
> > > As illustrated in Figure 2, the outputs of GRASS are the warmed-up matrix and the adjacency matrix. For graph-structure-agnostic models, this warmed-up matrix acts as the initial feature matrix. Consequently, methods like Mean and GAIN can benefit from using this enhanced matrix rather than relying solely on the initial matrix with missing features. Additionally, GRAPE, which constructs a heterogeneous GNN, can utilize more edges than before, thereby benefiting from the enhanced training. Furthermore, IGRM, an advanced version of GRAPE that also incorporates a sample-wise, i.e., row-wise, graph, has been initialized with the adjacency matrix produced by GRASS. This approach has demonstrated compatibility and improved performance, as evidenced in the tables mentioned above and in the medical datasets detailed in Appendix A.5.

---

> > > > ### Comment · Reviewer_T1VM · 2023-11-23
> > > >
> > > > I thank the authors for the response. All of my questions have been answered.
> > > >
> > > > However, the concern about hyperparameter setup remains. GCN models used in [1] are not a part of FP but one choice among downstream GNNs. Additionally, the datasets' domains differ between [1] and this paper. Could the use of a hidden dimension of 64 in this paper be justified solely because the hidden dimension of 64 is used in [1]? If the hidden dimension is changed to 16 or 256, could there be different performance trends among the methods? Moreover, wouldn't the optimal hidden dimension vary depending on the methods?
> > > >
> > > > I will raise my score and I am open to raising it even further if the aforementioned concern is addressed.
> > > >
> > > > [1] Rossi, Emanuele, et al. "On the unreasonable effectiveness of feature propagation in learning on graphs with missing node features." Learning on Graphs Conference. PMLR, 2022.

---

> ### Author Response · Authors · 2023-11-23
> **Additional Response to Reviewer T1VM**
>
> We are immensely grateful for your recognition of our rebuttal, particularly the discussion on convergence with Reviewer **T1VM**, which we found truly inspiring and significant from our perspective as authors. To address your remaining concerns, we have broadened our experiments to illustrate the impact of hyperparameters, encompassing not only the hidden dimensions in range {16, 64, 256} but also the dropout rates in range {0.0, 0.25, 0.5}, as initially pointed out in [Cons 2.] by the reviewer. We will begin by presenting the experimental results, followed by a summary of the key findings. For dataset, we used Pancreas dataset in bio domain with model variants in GCNMF (graph-perspective), Mean (tabular-perspective), scFP (domain-perspective).
>
> |                                   |    GCNMF    |  + GRASS init. |     Impr. (%)    |
> |:---------------------------------:|:-----------:|:-------------------:|:----------------:|
> |   Hidden dim: 16 / Dropout: 0.0   | 0.173+0.129 |     0.144+0.100     |       -16.76 |
> |   Hidden dim: 16 / Dropout: 0.25  | 0.058+0.007 |     0.329+0.252     |       467.24 |
> |   Hidden dim: 16 / Dropout: 0.5   | 0.098+0.054 |     0.340+0.296     |       246.94 |
> |   Hidden dim: 64 / Dropout: 0.0   | 0.671+0.056 |     0.680+0.067     |        1.34  |
> |   Hidden dim: 64 / Dropout: 0.25  | 0.360+0.247 |     0.662+0.048     |        83.89 |
> |   Hidden dim: 64 / Dropout: 0.5   | 0.443+0.026 |     0.677+0.065     |        52.82 |
> |   Hidden dim: 256 / Dropout: 0.0  | 0.603+0.008 |     0.671+0.053     |        11.28 |
> | **Hidden dim: 256 / Dropout:   0.25** | 0.611+0.009 |     **0.692**+0.049     |        13.26 |
> |   Hidden dim: 256 / Dropout: 0.5  | 0.617+0.023 |     0.667+0.051     |        8.10  |
>
> |                                   |     Mean    | + GRASS init. |    Impr. (%)    |
> |:---------------------------------:|:-----------:|:------------------:|:---------------:|
> |   Hidden dim: 16 / Dropout: 0.0   | 0.564+0.004 |     0.627+0.044    |      11.17      |
> |   Hidden dim: 16 / Dropout: 0.25  | 0.564+0.004 |     0.636+0.048    |       12.77 |
> |   Hidden dim: 16 / Dropout: 0.5   | 0.564+0.004 |     0.613+0.026    |        8.69 |
> |   Hidden dim: 64 / Dropout: 0.0   | 0.564+0.004 |     0.617+0.014    |        9.40 |
> |   Hidden dim: 64 / Dropout: 0.25  | 0.564+0.004 |     0.617+0.014    |        9.40 |
> |   Hidden dim: 64 / Dropout: 0.5   | 0.564+0.004 |     0.624+0.039    |       10.64 |
> |   Hidden dim: 256 / Dropout: 0.0  | 0.564+0.004 |     0.639+0.056    |       13.30 |
> | Hidden dim: 256 / Dropout:   0.25 | 0.564+0.004 |     0.638+0.052    |       13.12 |
> |   **Hidden dim: 256 / Dropout: 0.5**  | 0.564+0.004 |     **0.640**+0.064    |       13.48 |
>
>
> |                                   |     scFP    | + GRASS init. |    Impr. (%)   |
> |:---------------------------------:|:-----------:|:------------------:|:--------------:|
> |   Hidden dim: 16 / Dropout: 0.0   | 0.748+0.055 |     0.764+0.056    |       2.14 |
> |   Hidden dim: 16 / Dropout: 0.25  | 0.748+0.055 |     0.764+0.059    |       2.14 |
> |   Hidden dim: 16 / Dropout: 0.5   | 0.748+0.055 |     0.786+0.080    |       5.08 |
> |   Hidden dim: 64 / Dropout: 0.0   | 0.748+0.055 |     0.799+0.095    |       6.82 |
> |   Hidden dim: 64 / Dropout: 0.25  | 0.748+0.055 |     0.798+0.095    |       6.68 |
> |   Hidden dim: 64 / Dropout: 0.5   | 0.748+0.055 |     0.799+0.095    |       6.82 |
> |   **Hidden dim: 256 / Dropout: 0.0**  | 0.748+0.055 |     **0.800**+0.094    |       6.95 |
> | Hidden dim: 256 / Dropout:   0.25 | 0.748+0.055 |     0.799+0.094    |       6.82 |
> |   Hidden dim: 256 / Dropout: 0.5  | 0.748+0.055 |     0.799+0.095    |       6.82 |
>
> In the GCNMF Table, it reveals that a hidden dimension size of 256 with a dropout rate of 0.25 is more advantageous than the conventional setting of a 64-dimensional hidden layer with a 0.5 dropout rate. Moreover, in the Table of Mean and scFP, the case of an imputation model equipped solely with a logistic classifier (i.e., a 1-layer MLP) and thus invariant to hyperparameter flexibility, we found performance gains when using our proposed method, GRASS, as an initializer. This improvement is attributed to the trainable parameters in a 2-layer MLP and the incorporation of feature gradient. In summary, while setting constant hyperparameters ensures reproducibility and consistency across different datasets in the biomedical domain, allowing flexibility in these parameters can potentially enhance overall performance, as the reviewer suggested. For our finalized version, we would incorporate more flexibility, such as the number of layers.
>
> We have also created a new section in the updated paper to facilitate this discussion, which you can find in Appendix A.9, on Page 26.
>
> Once again, the enhancements made to this paper were possible due to the reviewer's invaluable feedback.
> We hope our latest efforts will further alleviate any remaining concerns.

---

> > ### Author Response · Authors · 2023-11-23
> > **Gentle Reminder for Reviewer T1VM**
> >
> > This is a gentle last-minute reminder regarding the rebuttal period. We have diligently addressed the concerns raised by the reviewers, as outlined above, and are hopeful for a positive response in the final stage of evaluating our work, hoping that we have appropriately addressed your first 8 concerns with the last mentioned concern.
> >
> > For your reference, we have included additional experiments in our revised manuscript, focusing on the key factors contributing to the performance boost from GRASS. These details can be found in Appendix A.10, on Page 26, with all the updated contents.
> >
> > We sincerely appreciate the invaluable time you have dedicated to this process.
> >
> > Best regards,
> > Authors

---

> > > ### Comment · Reviewer_T1VM · 2023-11-23
> > >
> > > I appreciate the authors for addressing most of my concerns. I changed my score to 6.

---

### Official Review · Reviewer_JSNu · 2023-11-01

**Soundness:** 3 good
**Presentation:** 4 excellent
**Contribution:** 3 good
**Rating:** 6
**Confidence:** 4

**Summary:**

Graph-based imputation methods typically depend on the presence of a graph structure. In the biomedical domain, an inherent graph structure among samples is often absent. This paper introduces GRASS, designed to augment the feature matrix and unearth the graph structure. GRASS utilizes feature gradients as an addition and engages in feature-wise propagation to produce an expanded feature matrix. Subsequently, it identifies the graph structure using sample-wise k-NN. The enhanced feature matrix combined with the newly identified graph structure can be applied to succeeding graph-based imputation techniques.

**Strengths:**

- The motivation is clearly explained.
- The use of feature gradients as a supplement seems novel.
- valuations span multiple datasets, backed by extensive analyses.
- The paper is well-structured and reader-friendly.
- The corresponding code has been made publicly accessible.

**Weaknesses:**

- The datasets in the study are somewhat limited, with a maximum of 8K samples. It remains to be seen how GRASS will perform on larger datasets. Will an increased sample count obstruct the column-wise feature propagation?

- Merging GRASS with existing methods adds to computational demands. The performance of current techniques with computational costs equivalent to GRASS's is not clear.

- Part two of section 4.3 is confusing to me. Figure 4 (b) appears to indicate similar gradient magnitudes for four genes, which doesn't necessarily suggest uniform gradient patterns among them. The authors also point out that gradients undergo normalization before being appended to the primary feature matrix. Hence, the gradient norm in Figure 4 (b) might be less indiactive. Perhaps pairwise cosine similarity could offer a more insightful metric.

** I am willing to raise the score if the authors can address the questions above.

**Questions:**

Refer to the aforementioned weaknesses for key questions, and also consider:
- What's the reasoning behind a 10% training, 10% validation, and 80% testing data split? Why is only 10% of the data used for training?
- Can you provde more descriptions regarding the baseline methods?
- Can you elaborate more on the datasets and the preprocessing steps? For instance, how were samples chosen? What features were considered? What targets were set for prediction?

---

> ### Author Response · Authors · 2023-11-22
> **Response to Reviewer JSNu (1/3)**
>
> We extend our sincere thanks to Reviewer JSNu for their time and constructive comments. We are eager to engage in further discussion with Reviewer JSNu and hope that our response effectively addresses their concerns.
>
> **[Cons 1: Limited Dataset with Larger Samples]**
> We acknowledge and appreciate the reviewer’s concern regarding the scalability of our algorithm to larger datasets. In response, we have expanded our testing to include a substantially larger dataset from the scRNA-seq domain, known as the Macosko dataset. This dataset is notably extensive, encompassing 44,808 cells, 22,452 genes, and 14 distinct cell types, with an initial missing rate of 81.41%. For our analysis, we preprocessed 2,000 highly variable genes, a standard practice in scRNA-seq studies. The performance results obtained from this dataset are summarized below.
>
>
> |          | Macosko (IMR: 81.41%) |             |             |
> |----------|:---------------------:|:-----------:|:-----------:|
> |          |           OG          | GRASS init. | Impr. (%) |
> | LP       |      0.853+0.025      | 0.870+0.025 |  1.99  |
> | GCNMF    |          OOM          |      -      |      -      |
> | PaGNN    |      0.938+0.008      | 0.939+0.001 |     0.17    |
> | GCN-zero |      0.920+0.031      | 0.929+0.006 |     0.92    |
> | GCN-nm   |      0.923+0.017      | 0.930+0.071 |     0.71    |
> | GCN-FP   |      0.937+0.006      | 0.941+0.045 |     0.43    |
> | PCFI     |      0.932+0.017      | 0.939+0.005 |     0.75    |
> | Mean     |      0.819+0.042      | 0.835+0.048 |     1.87    |
> | kNN      |      0.904+0.021      | 0.910+0.012 |     0.62    |
> | GAIN     |      0.891+0.039      | 0.898+0.012 |     0.86    |
> | MIWAE    |          OOM          |      -      |      -      |
> | GRAPE    |          OOM          |      -      |      -      |
> | IGRM     |          OOM          |      -      |      -      |
> | scFP     |      0.934+0.011      | 0.941+0.020 |     0.72    |
>
>
> We acknowledge that GRASS integrates effectively with existing methods, except in situations where initial baselines, such as GCNMF and GRAPE, face Out-Of-Memory (OOM) issues. These issues arise due to the weights associated with the Gaussian Mixture Model in GCNMF and the construction of a heterogeneous node-feature graph in GRAPE. However, it is observed that graph-based methods generally enhance their performance when integrated with GRASS. In large graphs, where the number of samples typically exceeds the feature dimension, GRASS aligns well with current graph-based imputation methods, owing to its computational efficiency.
>
> To summarize, our algorithm demonstrates good alignment with larger datasets due to its modest computational demands, which we will discuss further.
>
> **[Cons 2: Computational Demand]**
> The reviewer’s concern about computational demand alongside performance benefits is valid and important. We address this from two perspectives:
>
> ***Memory Cost***: The primary resource used by GRASS is the feature gradient ($\overline{\boldsymbol{\nabla}}_{\mathbf{X}} \in \mathbb{R}^{N \times F}$), which is instrumental in constructing a column-wise graph. This gradient has the same dimensions as the original feature matrix ($N$ nodes and $F$ features). In graph-based imputation models, the complexity of the row-wise adjacency matrix ($\mathbf{A} \in \mathbb{R}^{N \times N}$) often surpasses that of the feature matrix, resulting in a complexity of $\mathcal{O}(N^2)$, especially in biomedical datasets where $N >> F$. Therefore, the extra memory needed for the feature gradient is not excessively large. Moreover, the complexity of the generated column-wise graph ($\mathbf{A}^{feat} \in \mathbb{R}^{F \times F}$) is lower than that of the row-wise adjacency matrix, allowing GRASS to integrate with existing graph-based models without significant memory costs. Post computation of the warmed-up matrix ($\hat{\mathbf{X}}$) and adjacency matrix ($\hat{\mathbf{A}}$), the memory allocated for the feature gradient and column-wise graph can be freed, leaving only the training cost of the baseline model.
>
> ***Time Cost***: The time complexity of GRASS is comparable to training a conventional 2-layer MLP, efficient for tabular data. This involves training two weight matrices – one for the raw feature space to hidden space transformation, and another for mapping the hidden space to output space. The computation of the feature gradient, despite its apparent complexity as outlined in Proposition 1, is straightforward. The `requires_grad` switch enables automatic saving of gradient information, equating the time complexity for computing the feature gradient to that of training a 2-layer MLP. Moreover, column-wise Feature Propagation can be executed efficiently via sparse multiplication, ensuring that the time required for obtaining the warmed-up matrix and adjacency matrix is not significant.

---

> ### Author Response · Authors · 2023-11-22
> **Response to Reviewer JSNu (2/3)**
>
> **[Cons 3: Gradient Norm in Figure (4) and Pairwise Cosine Similarity]**
> We are grateful to the reviewer for their insightful feedback, which significantly contributes to the improvement of our paper. We agree with the reviewer's suggestion that normalizing gradients may render the gradient norm less indicative. The original intention behind Figure 4(b) was to showcase the gradient dynamics of key marker genes. However, upon reflection and in line with the reviewer's suggestion, we find that presenting the pairwise cosine similarity between the original input feature matrix and the feature gradients offers a more intuitive and insightful perspective. Our analysis of this similarity has yielded promising results, as detailed below.
>
>
> | $\mathbf{X}$    | COL1A1 | TIMP1 | TGFBI | PDGFRB |
> |--------|--------|-------|-------|--------|
> | COL1A1 | 1.00   | 0.66  | 0.50  | 0.14   |
> | TIMP1  | 0.66   | 1.00  | 0.48  | 0.11   |
> | TGFBI  | 0.50   | 0.48  | 1.00  | 0.10   |
> | PDGFRB | 0.14   | 0.11  | 0.10  | 1.00   |
>
> | $\overline{\boldsymbol{\nabla}}_\mathbf{X}$ | COL1A1 | TIMP1 | TGFBI | PDGFRB |
> |------------|--------|-------|-------|--------|
> | COL1A1     | 1.00   | 0.63  | 0.70  | 0.81   |
> | TIMP1      | 0.66   | 1.00  | 0.52  | 0.60   |
> | TGFBI      | 0.70   | 0.52  | 1.00  | 0.66   |
> | PDGFRB     | 0.81   | 0.60  | 0.66  | 1.00   |
>
> Here, it shows the marker genes for `activated stellate' such as COL1A1, TIMP1, TGFBI, and PDGFRB exhibit higher cosine similarity in feature gradient compared to its original matrix. This is achieved by incorporating task-relevant information, brought from label supervision, into the feature gradient. Following the reviewer's valuable suggestion, we have updated Figure 4(b) with a clearer representation to better illustrate this point. We kindly invite the reviewer to review this new version in the updated PDF file.
>
> (link to Figure 4: https://anonymous.4open.science/r/grass-iclr-41D5/figs/crop_confusion.pdf)
>
> **[Question 1: Reasoning Behind 10/10/80 Split]**
> Our choice of a 10/10/80 split was designed to mirror real-world scenarios, where training labels often form a small subset. This experimental setting aimed to evaluate the suitability of our proposed method in environments with limited training labels. However, we acknowledge the reviewer's suggestion that a larger training dataset could be beneficial. In such scenarios, we anticipate that GRASS would gain further from more substantial supervision, thereby enhancing the knowledge content in the feature gradient. We plan to explore and include a variety of training settings in our final version for a more comprehensive validation.
>
>
> **[Question 2. Descriptions regarding the baselines & Preprocessing of dataset]**
> We apologize for previously omitting detailed descriptions of baselines and preprocessing methods for the bio and medical datasets. In response to the reviewer's request, we provide the following additional information:
>
> In our bio datasets, we utilize cell-gene matrices to predict relevant annotated cell types for each cell. This cell type information serves as a supervisory signal during training. For preprocessing, we filter out cells and genes that have not been transcribed in each row and column, respectively, and apply a log transformation to normalize the count values.

---

> > ### Author Response · Authors · 2023-11-22
> > **Response to Reviewer JSNu (3/3)**
> >
> > - **The Mouse ES** dataset employs a droplet-microfluidic approach for parallel barcoding. We use concatenated data, originally separated by different days post-leukemia inhibitory factor (LIF) withdrawal, treating the day of withdrawal as the annotation for the cell type.
> >
> > - **The Pancreas** dataset, obtained via the inDrop method, captures the transcriptomes of individual pancreatic cells from four human donors and two mouse strains. It includes 14 distinct annotated cell types.
> >
> > - **The Baron Human** dataset focuses on individual pancreatic cells from human donors, sequenced using a droplet-based method. It also features 14 annotated cell types.
> >
> > - **The Mouse Bladder** dataset, sourced from the Mouse Cell Atlas (MCA) project and sequenced via the Microwell-seq platform, includes cell types as defined by the original authors' annotations.
> >
> > In our medical datasets, we focus on datasets that originally include missing values and feature a mix of categorical and numerical features. During preprocessing, we remove rows and columns if all features are missing in each sample or if all samples are missing in each feature, respectively. We select the most representative feature column related to the patient's diagnosis as the class label for prediction.
> >
> > - **Breast Cancer**: Published in the UCI repository and provided by the Oncology Institute, this dataset contains tumor-related features. We use 'recurrence', a binary attribute, as the class label.
> >
> > - **Hepatitis**: Also from the UCI repository, this dataset includes data on hepatitis occurrences in individuals, with attributes related to liver characteristics. The binary annotation of the patient's outcome (die or live) serves as the class label.
> >
> > - **Duke Breast**: Available from The Cancer Imaging Archive (TCIA), this dataset consists of medical images and non-image clinical data for tumor prediction. We utilize the 'Tumor\_Grade' feature from the tabular data, indicating the grade of the tumor, as the class label.
> >
> > - **ADNI**: This collection includes various types of medical images and non-image clinical data related to Alzheimer's disease. The 'DX\_bl' feature from the clinical data, indicating the patient's diagnosis, is used as the class label.
> >
> > - **ABIDE**: Containing data on autism spectrum disorder from brain imaging and clinical data, this dataset uses the 'DX\_Group' feature from the clinical data, representing the diagnostic group of the patient, as the class label.
> >
> > To address the challenge of generalizing graph-based imputation methods to tabular data, we have adopted two types of baseline approaches. For graph-based imputation methods, typically targeting downstream tasks like classification, we have utilized widely-used methods as follows. For GNN-based methods, a 2-layer GCN serves as a classifier.
> >
> > - **LP**, a semi-supervised algorithm, spreads known labels to similar data points in an unlabeled dataset, based on the given graph structure.
> > - **GCNMF**, an end-to-end GNN-based model, imputes missing features by assuming a Gaussian Mixture Model aligned with GCN.
> > - **PaGNN**, a GNN-based method, implements a partial message-passing scheme, propagating only observed features.
> > - **GCN-Zero**, a simple 2-layer Graph Convolution Network, imputes missing features with zeros.
> > - **GCN-NM**, imputes missing features by averaging the features of one-hop neighboring nodes, followed by GCN layers.
> > - **GCN-FP**, propagates given features through neighbors, replacing observed ones with their original values to minimize Dirichlet energy.
> > - **PFCI** improves upon FP by considering propagation among feature channels with pseudo confidence, defined by the shortest path to the known feature.
> >
> > Additionally, our primary focus being on tabular data, we include common table-based imputation methods as follows:
> >
> > - **Mean**, replaces missing values in a dataset with the mean value of the available data for the same feature.
> > - **kNN**, imputes missing data by finding the k nearest neighbors based on cosine similarity and then averaging their features.
> > - **GAIN**, a generative adversarial network, imputes missing values where one network generates candidates and another evaluates them.
> > - **MIWAE**, employing a type of autoencoder, offers multiple imputations, capturing the data's underlying distribution to provide multiple plausible values for missing data.
> > - **GRAPE**, adopting a bipartite graph framework, views observations and features as two node types and imputes missing values through edge-level prediction.
> > - **IGRM**, enhances the bipartite graph framework by introducing the concept of a friend network, denoting relationships between samples.
> >
> > For a detailed summary and related citations, we kindly direct the reviewer to Appendix A.3, on Page 18 of our manuscript.

---

> > > ### Author Response · Authors · 2023-11-23
> > > **Gentle Reminder for Reviewer JSNu**
> > >
> > > This is a gentle last-minute reminder regarding the rebuttal period. We have diligently addressed the concerns raised by the reviewers, as outlined above, and are hopeful for a positive response in the final stage of evaluating our work hoping that we have appropriately addressed your aforementioned concerns.
> > >
> > > For your reference, we have included additional experiments in our revised manuscript, focusing on the key factors contributing to the performance boost from GRASS. These details can be found in Appendix A.10, on Page 26, with all the updated contents.
> > >
> > > We sincerely appreciate the invaluable time you have dedicated to this process.
> > >
> > > Best regards,
> > > Authors

---

### Official Review · Reviewer_Cqu6 · 2023-11-01

**Soundness:** 3 good
**Presentation:** 2 fair
**Contribution:** 3 good
**Rating:** 6
**Confidence:** 3

**Summary:**

The paper proposes a novel method for handling missing data in tabular datasets. It leverages feature gradients to help construct a graph and use column-wise feature propagation to impute missing values based on their similarity to other features. This approach allows for effective imputation of missing values, and it is demonstrated to be effective and generalizable in handling various missing data scenarios, particularly in the context of medical and bio-domain datasets.

**Strengths:**

- Exploring the implicit graph structure among tabular data are useful for imputing the missing values.

- The paper leverages an interesting observation that feature gradients are more discriminative than the original input to distinguish classes. This insight highlights the value of utilizing these gradients in the context of data imputation, as they provide a more discriminative representation of the data.

- Extensive experiments are conducted on multiple datasets to show the improvement of the method.

**Weaknesses:**

- The gradients are normalized into unit vectors. However, the GCN propagation in equation 2 does not take the scales of different features into consideration. The graph propagation can be viewed as a weighted sum among columns. Some columns in greater magnitude may lead to large imputed values. Is this issue addressed in the proposed method?

- The definition and proposition is a bit redundant. The detail expression of the gradient is not used anywhere in the paper. Instead it will be more helpful to add a piece of pseudo code of algorithm to help the reader understand the whole framework.

- Is the choice of threshold theta column-dependent? It might not be the optimal to use a constant across all features. Theta might depends on the distribution of 0/1 in that feature.

**Questions:**

Please see weakness.

---

> ### Author Response · Authors · 2023-11-22
> **Response to Reviewer Cqu6**
>
> We deeply appreciate your recognition of the core intuition behind our paper, particularly regarding the role of feature gradients in aiding the discriminative representation of data. We hope that the following responses adequately address your concerns.
>
> **[Cons 1: Different Scales in GCN Propagation]**
> Thank you for highlighting the importance of normalization in the context of feature matrix scaling. Indeed, as you pointed out, the scale of imputed values, interpreted as weighted sums of columns, is crucial. To address this, we preprocess the feature matrix before its multiplication with the adjacency matrix. For numerical features, we employed a MinMaxScaler, ensuring a range of [0,1], as shown in line 163 of our preprocess.py script on our anonymous GitHub repository (https://anonymous.4open.science/r/grass-iclr-41D5/preprocess.py). Categorical features are one-hot encoded to align with the same range as numerical columns, as demonstrated in line 187 of the same script. This preprocessing ensures that the feature matrix values fall within [0,1], preventing the explosion of imputed values, considering the symmetrically normalized adjacency matrix with off-diagonal elements also ranging from [0,1]. This approach effectively addresses scalability issues in our algorithm.
>
> **[Cons 2: Expression of Gradient and Pseudocode of Algorithm]**
> We are grateful for your constructive feedback on expressing the concept of the feature gradient and the need for a clear pseudocode of our algorithm. The feature gradient, as defined in Definition 3.1 and Proposition 1, is the partial derivative of the loss with respect to the feature matrix, with its value dependent on trainable weight parameters related to the downstream task. This gradient informs the construction of a column-wise graph embedded with task-relevant information, distinguishing our method from existing approaches. The complete pseudocode, encompassing the process of obtaining the feature gradient, column-wise feature propagation, clamping technique, and final output, is provided in Appendix A.4 on Page 20. Additionally, for a more implementation-focused understanding, we have included a PyTorch-style implementation for obtaining the feature gradient in Appendix A.4 on Page 21, achieved simply by enabling `requires_grad=True`. We respectfully request the reviewer to consult this PyTorch-style pseudocode for a more in-depth understanding.
>
> **[Cons 3: Choice of Threshold $\theta$]**
> The clamping technique is applied across all categorical features using a global threshold $\theta$. We acknowledge that different thresholds could be significant for distinct features, especially in reflecting the inherent uncertainty in imputing categorical values, as addressed in Equation (3). The softmax operation is applied sample-wise for each categorical feature set. For example, in a one-hot encoded patient race category with three options (European, American, Japanese), imputed values of 0.8, 0.9, and 0.4 would transform to probabilities of 0.377, 0.416, and 0.207, respectively, post-softmax. A threshold of 0.4 would then yield a clamped output of [0,1,0] or [?,?,?] if the threshold is over 0.5. This allows subsequent imputation methods to fill in the gaps based on their algorithms. However, in cases with a large number of possible values for a category, the probability distribution might be too spread, as you rightly pointed out. Introducing a temperature parameter in the softmax operation could be a potential solution in such scenarios. We will thoroughly explore this aspect in our final version and sincerely thank you for this valuable insight.

---

> > ### Author Response · Authors · 2023-11-23
> > **Gentle reminder for Reviewer Cqu6**
> >
> > This is a gentle last-minute reminder regarding the rebuttal period. We have diligently addressed the concerns raised by the reviewers, as outlined above, and are hopeful for a positive response in the final stage of evaluating our work.
> >
> > For your reference, we have included additional experiments in our revised manuscript, focusing on the key factors contributing to the performance boost from GRASS. These details can be found in Appendix A.10, on Page 26, with all the updated contents.
> >
> > We sincerely appreciate the invaluable time you have dedicated to this process.
> >
> > Best regards,
> > Authors

---

### Official Review · Reviewer_r66i · 2023-11-08

**Soundness:** 2 fair
**Presentation:** 1 poor
**Contribution:** 2 fair
**Rating:** 5
**Confidence:** 5

**Summary:**

The authors present GRASS, an imputing method for tabular data that bridges the gap between graph-based imputation methods and real-world scenarios.
More precisely, GRASS employs a single-layer Multi-Layer Perceptron (MLP) in order to extract predictors gradient, which then serves as a resource for generating graph structures. After calculating these gradients, a graph from a feature (i.e., column) perspective is constructed and column-wise feature propagation is performed. Consequently, after the feature matrix is imputed, a second graph is generated but this time from a sample-oriented (i.e., row) perspective, which serves as the input for existing graph-based imputation models.
The authors present results for both single cell sequencing and medical datasets and compare their method to the state-of-the-art-imputation methods.

**Strengths:**

The authors introduce the concept of feature (predictors) gradient which is the partial derivative of the loss function for each of these features, in order to employ information both in a column- and row-wise manner. Via doing so they can estimate the salience of individual features in loss minimization. Through employing these gradients, the constructed graph encapsulates not just observed feature information from an initial state but also the feature saliency in relation to the targeted downstream task.
Their method seems to perform well also in cases where there is high missingness rate in the data.

**Weaknesses:**

The improvement induced by GRASS seems to be incremental and not consistent, especially given the computational trade-off (of calculating the feature gradients).
The presentation of the results needs to be improved -- it is unclear what they represent
Also there are some inconsistencies with respect to notation (eg Y in page 5 is not defines though used in the equation)

In terms of minor comments:
-- terms are defined more than once (eg. Feature Propagation (FP) (Rossi et al., 2021) )
-- Little et al., reference is presented as two distinct citations (2019a, 2019b)

**Questions:**

page 4, footnote:
What is meant by when employing zero imputation "steering clear of presumptions associated with missing completely at random" ?

Figure 3: within each dataset, what does each bar represents?
in case that this is GRASS against each of the other methods, why does GRASS performance isn't stable across all comparisons?

Figure 5: When looking at the (very small - almost unreadable) labels we see that there are 14 classes, nevertheless at 5(c) much fewer are presented and not in a much better manner as compared to 5(a).  Could you please comment on that

Table 1: what do the numbers in each column represent?

Figure 6(a): what is calculated within each cell of the k_col by k_row matrices?

What is the algorithmic convergence? Does it seem to converge eg with respect to loss, as the iteration number increases?

What is the loss function employed? Is the algorithm stable with respect to different loss functions ?
Is the uncertainty of imputing values calculated?

---

> ### Author Response · Authors · 2023-11-22
> **Response to Reviewer r66i (1/5)**
>
> We sincerely appreciate the reviewer's dedicated time in reviewing our paper and acknowledging the use of feature gradients, specifically feature saliency, targeted toward the downstream task. We are greatly encouraged by the reviewer's insightful summary of our work. In response to Reviewer r66i's questions, we provide detailed answers below.
>
> **[Cons 1: Incremental and Consistency]**
> We apologize if our presentation inadvertently misled the reviewer's understanding of our work. We would like to clarify that our proposed method, GRASS, serves as an agnostic preprocessing step. It generates a warmed-up feature matrix and adjacency matrix that seamlessly integrate with existing graph-based imputation methods. This is illustrated in each performance baseline, where the additional colored bars in green and brown (symbolizing 'grass') denote improvements when GRASS is utilized as an initializer. The extent of performance gain varies with the initial missing rate of the dataset and the size of the training sample. This variation explains the relatively lower performance gain in medical datasets compared to bio datasets, given the smaller number of training samples in a 10/10/80 train/val/test split. Overall, Figure 3 highlights the improved generalizability of current graph-based imputation methods when equipped with GRASS as an initializer. In the Introduction, we observed that graph-based imputation can sometimes underperform classical tabular-based methods, as seen with the Mouse ES dataset in Figure 3. However, upon integrating GRASS, its performance significantly increases, demonstrating GRASS's effectiveness within current graph-based imputation methods.
>
> **[Cons 2: Computational Trade-off]**
> Regarding the reviewer's concern about computational trade-offs versus performance gains, the computational demands of GRASS can be broken down into two aspects:
>
> ***Memory Cost***: GRASS primarily uses the feature gradient ($\overline{\boldsymbol{\nabla}}_{\mathbf{X}} \in \mathbb{R}^{N \times F}$), which supplements the construction of a column-wise graph and shares the same dimensions as the original feature matrix ($N$ nodes and $F$ features). In the context of graph-based imputation models, which typically use a row-wise adjacency matrix ($\mathbf{A} \in \mathbb{R}^{N \times N}$), the complexity of the adjacency matrix often exceeds that of the feature matrix ($\mathcal{O}(NF) + \mathcal{O}(N^2) = \mathcal{O}(N^2)$). This is particularly pertinent in biomedical domains, where datasets are tabular and the number of samples ($N$) significantly surpasses the number of features ($F$). Therefore, the extra memory needed for the feature gradient is relatively modest. Moreover, the complexity of the generated column-wise graph ($\mathbf{A}^{feat} \in \mathbb{R}^{F \times F}$) is also lower than the row-wise adjacency matrix. Thus, GRASS aligns with existing graph-based models without excessive memory costs. After computing the warmed-up matrix ($\hat{\mathbf{X}}$) and adjacency matrix ($\hat{\mathbf{A}}$), memory allocated for the feature gradient and column-wise graph can be freed, leaving only the cost of training the original baseline model.
>
> ***Time Cost***: The process resembles training a conventional 2-layer MLP, efficient for tabular data and involving two weight matrices. The calculation of the feature gradient, as detailed in Proposition 1, is straightforward in practice. By enabling the `requires_grad` switch, gradient information is automatically saved, equating the time complexity for computing the feature gradient to training a 2-layer MLP. Furthermore, column-wise Feature Propagation is executed efficiently via sparse multiplication, as detailed in our methodology. Therefore, the time required to obtain the warmed-up matrix and adjacency matrix is not significant.
>
> In conclusion, considering the relatively low memory and time complexities, we assert that the performance gains achieved with GRASS are substantial and meaningful.
>
> **[Cons 3: Presentation of Results]**
> We apologize to the reviewer for any confusion caused by our initial presentation of the results. To provide a clearer understanding of how GRASS improves existing baselines, we have reformatted the results into a table, as shown below:

---

> ### Author Response · Authors · 2023-11-22
> **Response to Reviewer r66i (2/5)**
>
> |          | Mouse ES (IMR: 27.21%) |              |           | Pancreas (IMR: 56.65%) |             |           | Baron Human (IMR: 57.25%) |             |           | Mouse Bladder (IMR: 69.05%) |             |           |
> |----------|:----------------------:|:------------:|:---------:|:----------------------:|:-----------:|:---------:|:-------------------------:|:-----------:|:---------:|:---------------------------:|:-----------:|:---------:|
> |          |           OG           |  GRASS init. | Impr. (%) |           OG           | GRASS init. | Impr. (%) |             OG            | GRASS init. | Impr. (%) |              OG             | GRASS init. | Impr. (%) |
> | LP       |       0.878+0.005      | 0.9794+0.003 |   11.43   |       0.656+0.039      | 0.798+0.068 |   21.66   |        0.736+0.022        | 0.828+0.055 |   12.46   |         0.556+0.030         | 0.643+0.053 |   15.57   |
> | GCNMF    |       0.525+0.238      | 0.9729+0.008 |   85.31   |       0.527+0.210      | 0.708+0.087 |   34.27   |        0.350+0.130        | 0.817+0.066 |   133.30  |         0.300+0.182         | 0.701+0.042 |   133.90  |
> | PaGNN    |       0.899+0.072      | 0.9806+0.002 |    9.03   |       0.701+0.044      | 0.768+0.040 |    9.58   |        0.777+0.043        | 0.820+0.057 |    5.53   |         0.713+0.056         | 0.775+0.028 |    8.78   |
> | GCN-zero |       0.960+0.005      | 0.9823+0.004 |    2.30   |       0.687+0.066      | 0.783+0.062 |   14.02   |        0.812+0.030        | 0.842+0.049 |    3.71   |         0.712+0.015         | 0.768+0.031 |    7.83   |
> | GCN-nm   |       0.885+0.098      | 0.9823+0.004 |   10.99   |       0.679+0.047      | 0.788+0.068 |   16.09   |        0.758+0.045        | 0.801+0.084 |    5.71   |         0.721+0.050         | 0.775+0.030 |    7.38   |
> | GCN-FP   |       0.900+0.100      | 0.9825+0.005 |    9.17   |       0.716+0.046      | 0.788+0.068 |   10.08   |        0.789+0.039        | 0.802+0.084 |    1.61   |         0.686+0.048         | 0.772+0.036 |   12.48   |
> | PCFI     |       0.949+0.004      |  0.955+0.006 |    0.57   |       0.673+0.055      |  0.686+0.04 |    1.95   |        0.769+0.036        | 0.792+0.038 |    2.96   |         0.710+0.046         | 0.727+0.028 |    2.41   |
> | Mean     |       0.979+0.006      | 0.9799+0.004 |    0.08   |       0.616+0.044      | 0.619+0.032 |    0.44   |        0.672+0.010        | 0.694+0.023 |    3.41   |         0.555+0.074         | 0.569+0.062 |    2.39   |
> | kNN      |       0.969+0.011      |  0.977+0.005 |    0.83   |       0.652+0.047      | 0.706+0.048 |    8.34   |        0.746+0.048        | 0.760+0.053 |    1.82   |         0.587+0.038         | 0.674+0.059 |   14.82   |
> | GAIN     |       0.978+0.011      | 0.9826+0.007 |    0.39   |       0.638+0.075      | 0.738+0.024 |   15.66   |        0.728+0.041        | 0.745+0.033 |    2.39   |         0.585+0.030         | 0.649+0.038 |   10.84   |
> | MIWAE    |           OOM          |       -      |     -     |            OOM           |      -      |     -     |             OOM             |      -      |     -     |              OOM              |      -      |     -     |
> | GRAPE    |           OOM          |       -      |     -     |            OOM           |      -      |     -     |             OOM             |      -      |     -     |              OOM              |      -      |     -     |
> | IGRM     |           OOM          |       -      |     -     |            OOM           |      -      |     -     |             OOM             |      -      |     -     |              OOM              |      -      |     -     |
> | scFP     |       0.952+0.004      |  0.976+0.003 |    2.52   |       0.743+0.044      | 0.788+0.085 |    6.05  |        0.809+0.067        | 0.853+0.031 |    5.43   |         0.653+0.024         | 0.759+0.022 |    16.23   |

---

> ### Author Response · Authors · 2023-11-22
> **Response to Reviewer r66i (3/5)**
>
> For Medical datasets the overall performance is as below:
>
> |          | Breast Cancer (IMR: 0.35%) |             |           | Hepatitis (IMR: 5.67%) |             |           | Duke Breast (IMR: 11.94%) |             |           | ADNI (IMR: 30.02%) |             |           | ABIDE (IMR: 69.74%) |             |           |
> |----------|:--------------------------:|:-----------:|:---------:|:----------------------:|:-----------:|:---------:|:-------------------------:|:-----------:|:---------:|:------------------:|:-----------:|:---------:|:-------------------:|:-----------:|:---------:|
> |          |             OG             | GRASS init. | Impr. (%) |           OG           | GRASS init. | Impr. (%) |             OG            | GRASS init. | Impr. (%) |         OG         | GRASS init. | Impr. (%) |          OG         | GRASS init. | Impr. (%) |
> | LP       |         0.561+0.038        | 0.562+0.041 |    0.14   |       0.573+0.078      | 0.608+0.053 |    6.06   |        0.672+0.021        | 0.678+0.026 |    0.98   |     0.928+0.005    | 0.943+0.005 |    1.56   |     0.894+0.009     | 0.895+0.011 |    0.13   |
> | GCNMF    |         0.551+0.033        | 0.579+0.049 |    5.02   |       0.685+0.097      | 0.707+0.088 |    3.22   |        0.664+0.035        | 0.688+0.032 |    3.61   |     0.897+0.045    | 0.944+0.004 |    5.25   |     0.819+0.042     | 0.913+0.010 |   11.49   |
> | PaGNN    |         0.540+0.037        | 0.562+0.032 |    3.98   |       0.729+0.074      | 0.741+0.058 |    1.74   |        0.685+0.033        | 0.690+0.029 |    0.69   |     0.953+0.003    | 0.955+0.003 |    0.27   |     0.907+0.009     | 0.914+0.008 |    0.82   |
> | GCN-zero |         0.542+0.048        | 0.557+0.039 |    2.71   |       0.713+0.090      | 0.714+0.088 |    0.14   |        0.673+0.022        | 0.694+0.021 |    3.13   |     0.956+0.003    | 0.957+0.003 |    0.17   |     0.902+0.008     | 0.915+0.008 |    1.38   |
> | GCN-nm   |         0.538+0.049        | 0.566+0.052 |    5.07   |       0.702+0.071      | 0.702+0.071 |    0.00   |        0.678+0.033        | 0.691+0.025 |    1.96   |     0.955+0.003    | 0.956+0.003 |    0.19   |     0.905+0.011     | 0.918+0.007 |    1.48   |
> | GCN-FP   |         0.543+0.047        | 0.565+0.052 |    4.05   |       0.705+0.085      | 0.707+0.092 |    0.20   |        0.661+0.031        | 0.688+0.028 |    4.21   |     0.955+0.003    | 0.957+0.003 |    0.18   |     0.908+0.014     | 0.915+0.005 |    0.86   |
> | PCFI     |         0.545+0.039        | 0.547+0.040 |    0.44   |       0.728+0.108      | 0.728+0.108 |    0.00   |        0.693+0.029        | 0.696+0.030 |    0.40   |     0.951+0.004    | 0.955+0.003 |    0.46   |     0.915+0.008     | 0.917+0.010 |    0.26   |
> | Mean     |         0.562+0.045        | 0.562+0.045 |    0.00   |       0.691+0.072      | 0.711+0.081 |    2.86   |        0.687+0.018        | 0.687+0.019 |    0.04   |     0.939+0.002    | 0.943+0.003 |    0.46   |     0.607+0.027     | 0.905+0.007 |   49.09   |
> | kNN      |         0.552+0.041        | 0.556+0.041 |    0.67   |       0.612+0.097      | 0.626+0.105 |    2.15   |        0.692+0.026        | 0.697+0.014 |    0.74   |     0.943+0.003    | 0.943+0.004 |    0.01   |     0.896+0.009     | 0.907+0.010 |    1.16   |
> | GAIN     |         0.566+0.044        | 0.567+0.043 |    0.21   |       0.578+0.093      | 0.646+0.080 |   11.63   |        0.699+0.018        | 0.699+0.017 |    0.09   |     0.937+0.003    | 0.944+0.003 |    0.67   |     0.793+0.010     | 0.910+0.009 |   14.70   |
> | MIWAE    |         0.558+0.033        | 0.563+0.035 |    0.93   |       0.573+0.080      | 0.608+0.077 |    6.25   |        0.692+0.013        | 0.693+0.012 |    0.13   |          OOM         |      -      |     -     |     0.623+0.015     | 0.898+0.008 |   44.10   |
> | GRAPE    |         0.572+0.029        | 0.573+0.017 |    0.26   |       0.701+0.033      | 0.706+0.032 |    0.63   |            OOM            |      -      |     -     |          OOM         |      -      |     -     |     0.889+0.010     | 0.906+0.006 |    1.90   |
> | IGRM     |         0.548+0.039        | 0.552+0.037 |    0.66   |       0.668+0.087      | 0.703+0.109 |    5.26   |            OOM            |      -      |     -     |          OOM         |      -      |     -     |     0.747+0.019     | 0.908+0.004 |   21.54   |
> | scFP     |         0.554+0.047        | 0.563+0.055 |    1.62   |       0.691+0.077      | 0.691+0.077 |    0.00   |        0.678+0.031        | 0.690+0.030 |    1.76   |     0.953+0.003    | 0.954+0.002 |    0.10   |     0.894+0.010     | 0.903+0.007 |    1.00   |
>
> We have included a more visually detailed illustration of the table in the updated PDF, located in Appendix A.5 on page 22.
> We respectfully request the reviewer to consider this illustration in the Appendix for a comprehensive understanding.

---

> ### Author Response · Authors · 2023-11-22
> **Response to Reviewer r66i (4/5)**
>
> **[Cons 4: Inconsistencies in Notation]**
> We appreciate the reviewer's attention to detail regarding the notation. The symbol $ Y $ is clearly defined in Proposition 1 as a label matrix used during the computation of the feature gradient, and is denoted in orange. This specific use of $ Y $ is confined to the calculation of the feature gradient and does not recur elsewhere in the paper.
>
> **[Minor Comments]**
> Thank you for pointing out the redundancy in terms and the need for clarity in citations. Based on your valuable feedback, we have revised the redundant terms and clarified the citations in the updated PDF version.
>
> **[Question 1: “Clear of Presumptions Associated with Missing Completely at Random (MCAR)”]**
> We apologize for any ambiguity in our wording. By "Steering clear of presumptions associated with Missing Completely at Random (MCAR)," we mean that our method of zero imputation avoids assuming that the missing data occurred completely at random. This assumption may not always hold true in real-world datasets, making our approach more flexible and generalizable. This is especially pertinent in the biomedical domain, where zero often has a meaningful interpretation, such as the absence of a feature.
>
> **[Question 2: Meaning of Each Bar in Figure 3]**
> In Figure 3, each bar represents the performance (AUROC for medical datasets, Macro-F1 for bio datasets) of existing methods, as well as their enhanced performance when GRASS is used as an initializer.
>
> **[Question 3: Fewer Labels in Figure 5]**
> We are grateful to the reviewer for identifying the discrepancy in Figure 5 regarding the representation of cell-types. We have corrected the figure to accurately reflect all 14 cell-types. Recognizing the figure's small size, we have provided a link for a clearer view of the refined tSNE visualization. This figure demonstrates that, compared to the original feature matrix, the feature gradient imbued with cell-type classification knowledge significantly enhances the discernment of different cell-types in the warmed-up matrix through column-wise FP.
>
> (link to Figure 5: https://anonymous.4open.science/r/grass-iclr-41D5/figs/crop_tsne_final.pdf)
>
> **[Question 4: Meaning of Numbers in Each Column in Table 1]**
> Table 1, presenting our ablation study, contains cells that denote the average AUROC score along with the standard deviation across five runs. To facilitate understanding, we have explicitly mentioned this metric in the manuscript.
>
> **[Question 5: Meaning of Each Cell in Figure 6 (a)]**
> We apologize for not providing detailed explanations for Figure 6 (a). This figure aims to illustrate the sensitivity of GRASS in terms of selecting the number of neighbors for column-wise and row-wise graphs, assessed through the AUROC metric. This metric is also described in the figure's caption.

---

> ### Author Response · Authors · 2023-11-22
> **Response to Reviewer r66i (5/5)**
>
> **[Question 6: Meaning of Algorithmic Convergence]**
> We appreciate the mention of the convergence property discussed in our manuscript. The convergence concept primarily stems from the FP paper, which discusses the convergence of imputed values. A key difference between our work and FP is the nature of the graph structure—whether it is manually constructed via $ k $NN (as in GRASS) or initially given (as in FP). This distinction is critical as the strong connectivity of the graph, where every node is accessible from others, is a prerequisite for convergence in imputed values. In our approach, utilizing $ k $NN-based graph generation, convergence is not guaranteed, as noted by Reviewer T1VM. However, we argue that in scenarios with prevalent missing features, especially in tabular data formats requiring manual graph generation, convergence is not necessarily advantageous. Our rationale is further elaborated in Appendix A.2., on Page 16. Briefly, as demonstrated in the table below, the risk of oversmoothing (indicated by the MADGap, which measures the representational difference between remote and neighboring nodes) can detrimentally impact performance, even if convergence is achieved.
>
> | Mouse ES (Macro-F1) |    | k_row |       |       |       |   |          |
> |:-------------------:|:--:|:-----:|:-----:|:-----:|:-----:|---|----------|
> |                     |    | 1     | 5     | 10    | 50    |   | MADGap   |
> |        K_col        | 50  | 0.893 | 0.840 | 0.841 | 0.842 |   | 1.43E-06  |
> |                     | 10 | 0.910 | 0.983 | 0.923 | 0.950 |   |  0.00085 |
> |                     | 5 | 0.767 | 0.919 | 0.977 | 0.945 |   | 0.00341  |
> |                     | 1 | 0.836 | 0.962 | 0.964 | 0.946 |   | 0.00355 |
>
> | ABIDE (AUROC) |    | k_row |       |       |       |   |          |
> |:-------------:|:--:|:-----:|:-----:|:-----:|:-----:|---|----------|
> |               |    | 1     | 5     | 10    | 50    |   | MADGap   |
> |     K_col     | 50  | 0.905 | 0.908 | 0.906 | 0.902 |   | 0.00E+00  |
> |               | 10  | 0.889 | 0.881 | 0.888 | 0.895 |   | 0.00148  |
> |               | 5 | 0.894 | 0.894 | 0.901 | 0.904 |   |  0.01134 |
> |               | 1 | 0.911 | 0.904 | 0.910 | 0.905 |   | 0.01048 |
>
> **[Question 7: Employed Loss Function and Uncertainty of Imputed Values]**
> We appreciate the reviewer's inquiry regarding the loss function used in our study. Indeed, for the downstream classification task, we employed the CrossEntropy loss. While this work concentrates on classification, aligning with existing graph-based imputation methods, we acknowledge the reviewer's point about the potential of using different loss functions for various tasks, such as MSE loss, KL divergence loss, or self-supervised loss. We find this suggestion intriguing and consider it an excellent direction for future research. We plan to incorporate this aspect of versatility into the final version of our paper, exploring the generalizability of graph-based methods across diverse loss functions.
>
> Regarding the implementation, obtaining task-relevant feature gradients is feasible using AutoGrad. By setting `requires_grad=True`, we can efficiently compute these gradients, underscoring the flexibility of our approach.
>
> As for the uncertainty of imputed values, especially in categorical features, we recognize the significance of reflecting such uncertainty. For instance, in biological datasets, an imputed value of 0.5 for a tissue type, where '0' denotes epithelial and '1' denotes mesenchymal, would be biologically nonsensical. To address this, as detailed in Section 3.3, we implemented a clamping technique. This technique employs a softmax operation to sample categorical features based on their specific number. For example, in a one-hot encoded representation of a patient's race (European, American, Japanese), imputed values like 0.8, 0.9, and 0.4 would be transformed via softmax to 0.377, 0.416, and 0.207, respectively. Setting a threshold, such as 0.4, allows the clamping operation to output a definite category or leave room for ambiguity (e.g., [0,1,0] or [?,?,?]) when the threshold is higher than 0.5. This approach enables subsequent imputation methods to fill in the data based on their algorithms. The efficacy of this clamping process is demonstrated in Table 1 on Page 9, where incorporating the clamping technique significantly enhances overall performance.

---

> ### Comment · Reviewer_r66i · 2023-11-23
> **Final score**
>
> We thank the authors for their response
> Their feedback is appreciated, nevertheless my score shall stay the same
> The authors are highly encouraged to extend their promising work

---

> ### Author Response · Authors · 2023-11-23
> **Additional Response to Reviewr r66i**
>
> Thank you for the time spent evaluating our rebuttal and for recognizing the promise in our work. We have made our utmost efforts to thoroughly address the concerns raised, as detailed above. Here, we would like to further clarify the origin of our performance gain, providing a clearer response to Reviewer r66i's **[Cons 1.]**.
>
> Below, we further explore how the performance enhancement is achieved by initializing current graph-based imputation methods with GRASS. Specifically, using GRASS as an initializer provides a warmed-up matrix along with a refined adjacency matrix. We assessed this adjacency matrix($\hat{\mathbf{A}}$) in terms of the homophily ratio, a crucial inductive bias in the graph domain.
>
> | Edge Homophily |   $\mathbf{A}$   |  $\hat{\mathbf{A}}$ | Impr. (%) |
> |:--------------:|:------:|:------:|:---------:|
> |    Mouse ES    | 0.8591 | 0.9770 |   13.724  |
> |    Pancreas    | 0.9319 | 0.9579 |   2.790   |
> |   Baron Human  | 0.9557 | 0.9696 |   1.454   |
> |  Mouse Bladder | 0.5672 | 0.7559 |   33.269  |
> |  Breast Cancer | 0.6698 | 0.6701 |   0.045   |
> |    Hepatitis   | 0.7902 | 0.8035 |   1.683   |
> |   Duke Breast  | 0.6887 | 0.6915 |   0.407   |
> |      ADNI      | 0.7130 | 0.7336 |   2.889   |
> |      ABIDE     | 0.9142 | 0.9166 |   0.263   |
>
> As shown in the above table, employing GRASS to obtain refined adjacency matrix ($\hat{\mathbf{A}}$) leads to an improved edge homophily ratio, indicating an increase in edges connecting nodes with the same labels. This improvement is attributed to the incorporation of feature gradients, which introduce task-relevant information through supervision signals from training the MLP. Consequently, a more refined graph structure, enriched with task-relevant information, allows current graph-based imputation methods to further enhance their performance by smoothing the representation of their nearest neighbors. In summary, this improvement in the graph structure is a key factor contributing to the performance boost observed with GRASS.
>
> We have also included this experiment in Appendix A.10, on Page 26.
>
> Overall, we hope that these experimental results will provide the reviewer with additional clarity and be taken into consideration while making the final decision on our work.
> Thank you for your invaluable time.
>
> Best,
> Authors

---

### Author Response · Authors · 2023-11-22
**General Responses**

We proposed the novel framework, GRASS, designed to enhance the generalizability of current graph-based imputation methods within the bio-medical domain, providing a warmed-up matrix and adjacency matrix derived from feature gradient and column-wise feature propagation.

We express our profound gratitude to all reviewers for dedicating their time and effort to thoroughly evaluate our manuscript. We especially appreciate the positive acknowledgments regarding “interesting observation”, “clear motivation”, "highlighting the value of utilizing feature gradients", and "reader-friendly". The constructive feedback offered by the reviewers has been invaluable in enhancing the quality of our work. Below is a summary of the key updates made to our paper, in addition to the detailed point-by-point responses:

**[Main Paper]**

- The focus on the bio-medical domain has been more explicitly defined. (@Reviewer T1VM)
- Redundant citations have been removed. (@Reviewer r66i)
- The notation $Y$ in Proposition 1 has been updated. (@Reviewer r66i)
- The use of a weighted adjacency matrix has been clarified. (@Reviewer T1VM)
- Figure 4 now includes the cosine similarity distance of marker genes. (@Reviewer T1VM)
- Figure 5 has been revised with distinct color maps for different cell types. (@Reviewer r66i)
- The metrics used in Figure 6 and Table 1 have been clearly specified. (@Reviewer r66i)

**[Appendix]**

- Convergence Property discussion added in Appendix A.2. (@Reviewer T1VM, r66i)
- Additional experimental details for datasets and baselines in Appendix A.3. (@Reviewer JSNu)
- PyTorch-style pseudocode included in Appendix A.4. (@Reviewer 4SJf, Cqu6)
- Enhanced visualizations in Appendix A.5. (@Reviewer T1VM, r66i)
- Comparison with scFP outlined in Appendix A.6. (@Reviewer T1VM)
- Discussion on Computational Demand in Appendix A.7. (@Reviewer r66i, JSNu)
- Further Extensions discussed in Appendix A.8. (@Reviewer r66i, JSNu)
- Hyperparameter Flexibility discussed in Appendix A.9. (@Reviewer T1VM)
- Edge Homophily Improvement discussed in Appendix A.10.

For the revised/updated parts, we have marked them in orange color. We hope the responses provided below will address any remaining concerns from the reviewers. We are open to further questions and will be delighted to provide additional clarifications.

Once again, we extend sincere thanks to the reviewers for their insightful feedback and valuable time.

Best,
Authors

---

### Meta-Review · Area_Chair_Xd7v · 2023-12-11

**Metareview:**

The work discusses on data imputation using graph-based approaches, with a view on tabular data. The approach is built up of a combination of ideas followed up with some empirical analysis. The experiments are mostly done with standard online benchmarks, which does not bring all the desirable flavour that a deep biomedical application can do. On that, there are many imputation methods which are left out, and it is unclear how these proposed ideas compare to other imputation methods, including for instance those that try to learn a graph factorisation of the joint distribution for the features. That said, one cannot do it all, and the paper is well presented and achieves reasonable results. In the end, the submission as presented has not greatly excited any committee member.

**Justification For Why Not Higher Score:**

Interesting combination of ideas and their use for tabular data but perhaps with somewhat limited scope and comparisons.

**Justification For Why Not Lower Score:**

N/A

---

### Decision · Program_Chairs · 2024-01-16

Reject